# Propagation of hippocampal ripples to the neocortex by way of a subiculum-retrosplenial pathway

Noam Nitzan [1,8], Sam McKenzie[2,8], Prateep Beed[1], Daniel Fine English [2,7], Silvia Oldani[1,3], John J. Tukker [1,3], György Buzsáki [2,4✉] & Dietmar Schmitz [1,3,5,6✉]

Bouts of high frequency activity known as sharp wave ripples (SPW-Rs) facilitate communication between the hippocampus and neocortex. However, the paths and mechanisms by which SPW-Rs broadcast their content are not well understood. Due to its anatomical positioning, the granular retrosplenial cortex (gRSC) may be a bridge for this hippocampo-cortical dialogue. Using silicon probe recordings in awake, head-fixed mice, we show the existence of SPW-R analogues in gRSC and demonstrate their coupling to hippocampal SPW-Rs. gRSC neurons reliably distinguished different subclasses of hippocampal SPW-Rs according to ensemble activity patterns in CA1. We demonstrate that this coupling is brain state-dependent, and delineate a topographically-organized anatomical pathway via VGlut2-expressing, bursty neurons in the subiculum. Optogenetic stimulation or inhibition of bursty subicular cells induced or reduced responses in superficial gRSC, respectively. These results identify a specific path and underlying mechanisms by which the hippocampus can convey neuronal content to the neocortex during SPW-Rs.

[1] Charité-Universitätsmedizin Berlin, Corporate member of Freie Universität Berlin, Humboldt-Universität zu Berlin, and Berlin Institute of Health, Neuroscience Research Center, Berlin, Germany. [2] Neuroscience Institute and Department of Neurology New York University Langone Medical Center, New York, NY 10016, USA. [3] Center for Neurodegenerative Diseases (DZNE), Berlin, Germany. [4] Center for Neural Science, New York University, New York, NY 10016, USA. [5] Cluster of Excellence NeuroCure, Berlin, Germany. [6] Einstein Center for Neurosciences, Berlin, Germany. [7] Present address: School of Neuroscience, College of Science, Virginia Tech, VA 24061, USA. [8] These authors contributed equally: Noam Nitzan, Sam McKenzie. ✉email: gyorgy.buzsaki@nyumc.org; dietmar.schmitz@charite.de

Theoretical considerations[1–3] have motivated a dual-stage hypothesis for learning, whereby new associations are first stored within the rapidly changing synapses of the hippocampus and subsequently broadcast to the rest of the brain during high-frequency oscillatory bursts known as sharp wave ripples (SPW-Rs)[4,5]. SPW-Rs are thought to promote the long-term storage of memories in distributed cortical networks through the strengthening of intra-cortical connectivity. As predicted by this model, several studies have shown coordinated reactivation of awake activity patterns in the hippocampus and various cortical areas during SPW-Rs[6–9]. In further support of the physiological importance of SPW-Rs, disrupting these events during slow-wave sleep (SWS) or awake behavior slows learning[10–12].

To date, little is known about the anatomical pathways that support the flow of information from the hippocampus to the neocortex during SPW-Rs. Perhaps the best-studied associational areas are the entorhinal cortex and the medial prefrontal cortex (mPFC), which both receive direct hippocampal projections[13,14] and show neural firing rate changes during hippocampal SPW-Rs[8,15,16]. In contrast, other associational areas lack direct connectivity with the hippocampus proper, despite being coupled to SPW-Rs[17–20]. Since SPW-Rs are embedded within other cortical rhythms, such as delta waves and spindles[15,21], it has been difficult to isolate the contribution of hippocampal output from ongoing cortical dynamics.

We addressed these questions by focusing on the granular retrosplenial cortex (gRSC), a comparatively unexplored target of the dorsal hippocampus that is involved in spatial navigation, and context learning[22–29]. The gRSC is situated at the septal hippocampo-cortical interface and is strongly interconnected with a variety of subcortical, sensory and associational areas[30–33], rendering it a prime candidate for mediating processed hippocampal output.

Using in vivo and in vitro recordings in combination with optogenetic tools in different transgenic mouse lines, we identified a subpopulation of hippocampal output neurons that mediate ripple-associated responses and their downstream targets in the gRSC. We show that these projections are topographically organized and that they are both necessary and sufficient for cortical SPW-R evoked responses. We propose that these connections represent the first stage in a dorso-medial pathway supporting the propagation of ripple activity.

## Results

**Transient ripple (140–200 Hz) oscillations in the gRSC.** To investigate the flow of information from the hippocampus to the gRSC, we performed silicon probe recordings in the dorsal CA1 and subiculum and different layers of the gRSC in awake, head-fixed mice ($n = 8$ C57Bl/6 N). Mice were habituated for 5–7 days until they remained quiet for approximately 1 h (Methods).

Laminar recordings from the gRSC with silicon probes (Fig. 1a) during immobility revealed a transient, fast oscillatory pattern in the frequency range 140–200 Hz, which we refer to as "gRSC ripple" (Fig. 1b). gRSC ripple wave-triggered spectrograms showed that the largest amplitude gRSC ripple was localized to deeper recording sites, which were histologically confirmed to correspond to superficial layers (Fig. 1a–c and Supplementary Fig. 1; $n = 19$ sessions from six animals). gRSC ripple oscillations co-occurred with a lower frequency negative wave, which had its largest amplitude in superficial layers and its polarity reversed in deeper layers. This phase reversal resulted in a corresponding sink-source dipole in the current source density (CSD) map (Fig. 1d, e and Supplementary Fig. 1). Phase-amplitude coupling analysis confirmed a significant coupling between lower frequencies phase

(~10 Hz) reflecting negative waves and ripple band power, as well as power in higher frequencies reflecting associated spiking activity (Supplementary Fig. 1). The fast oscillatory events were relatively short-lasting ($26.10 \pm 0.14$ ms; median ± s.e.m), consisting of $5.02 \pm 0.02$ cycles (Supplementary Fig. 1c–f) and occurred at a low incidence ($0.23 \pm 0.03$ Hz).

Next, we quantified the relationship between neuronal spiking and LFP in the gRSC. We isolated single units, which were classified as putative pyramidal neurons or interneurons[34] (Supplementary Fig. 2). Pyramidal cells were further divided into deep and superficial layer neurons based on post-hoc reconstruction of probe tracks (Fig. 1a). Neuronal firing in gRSC was time-locked to both gRSC ripple events coupled with negative waves (Fig. 1f) and to individual ripple waves (Fig. 1g). Examination of phase relationship between spikes and LFP ripple waves indicated that a larger proportion of superficial units were significantly phase-locked to ripple events (37%, 19/51) than neurons in deep layers (19%, 35/180). Superficial neurons also showed stronger increase in firing rate during gRSC ripples (Fig. 1g, h). The spiking of putative interneurons lagged behind that of the ripple trough-locked pyramidal neurons by 20–30°[35] (Fig. 1g; $p = 0.02$; two-sided rank sum test). As hippocampal, entorhinal and neocortical ripples often co-occur[16,17], we next examined whether hippocampal SPW-Rs propagate multi-synaptically to gRSC.

**Propagation of hippocampal SPW-Rs into the gRSC.** LFP activity in superficial layers of gRSC in immobile mice was dominated by large negative polarity waves (mean amplitude: $-91.42 \pm 0.21$ µV; mean frequency $7.82 \pm 1.66$ Hz), likely reflecting synaptic inputs to neurons in layers 2 and 3 (L2/3) and, possibly, to the apical dendrites of layer 5 (L5) neurons. Some, but not all, of these waves coincided with SPW-Rs recorded in CA1 pyramidal cell layer or in the subiculum (Fig. 2a–c). As the incidence of hippocampal SPW-Rs was lower (CA1 SPW-Rs: $0.68 \pm 0.11$ Hz) than that of the large-amplitude waves in the superficial layers of gRSC, only a fraction of the cortical waves would be expected to be related to CA1 SPW-Rs. Hippocampal SPW-R – gRSC negative wave correlation peaked within 10 ms after CA1 SPW-R onset and was often preceded by an additional smaller peak by approximately 100 ms (Fig. 2c), implying that co-occurrences of these cross-regional events were embedded in other cortical rhythms[21,36]. Temporal analysis of the correlation between CA1 and gRSC ripple events indicated that the rate of gRSC ripples significantly peaked immediately following CA1 SPW-R onset (Fig. 2d). Moreover, hippocampal ripple power was positively correlated with ripple power in the gRSC, indicating that stronger hippocampal SPW-Rs are more likely to propagate to the gRSC. Likewise, the majority of gRSC ripple events were associated with an increased CA1 ripple power, whereas a fraction of low power ripples were not (Fig. 2e, f). In a subset of experiments where CA1, subiculum and superficial gRSC layers were all successfully targeted, the time lags of subicular and gRSC ripple power peaks relative to CA1 were computed. Consistent with the anatomical connectivity, subicular ripple probability peaked after that of CA1 (median $0.80 \pm 0.32$ ms), while gRSC ripples exhibited an even longer delay from CA1 ripples (median $4.76 \pm 0.34$ ms; Fig. 2g). CA1 ripple event-triggered CSD analysis demonstrated the presence of a superficial sink in gRSC (Fig. 2h, left), suggesting the existence of lamina-specific depolarization. When individual waves of CA1 ripples were used as a reference, both LFP ripples and ripple-related CSD sinks in superficial gRSC traces (Fig. 2h, right) were smaller compared to when local gRSC ripples waves were used as the reference (cf. Fig. 1d). CSD analysis assumes vertical conductivity, which was violated in this study due to the angle of probe placement (Fig. 2a). To

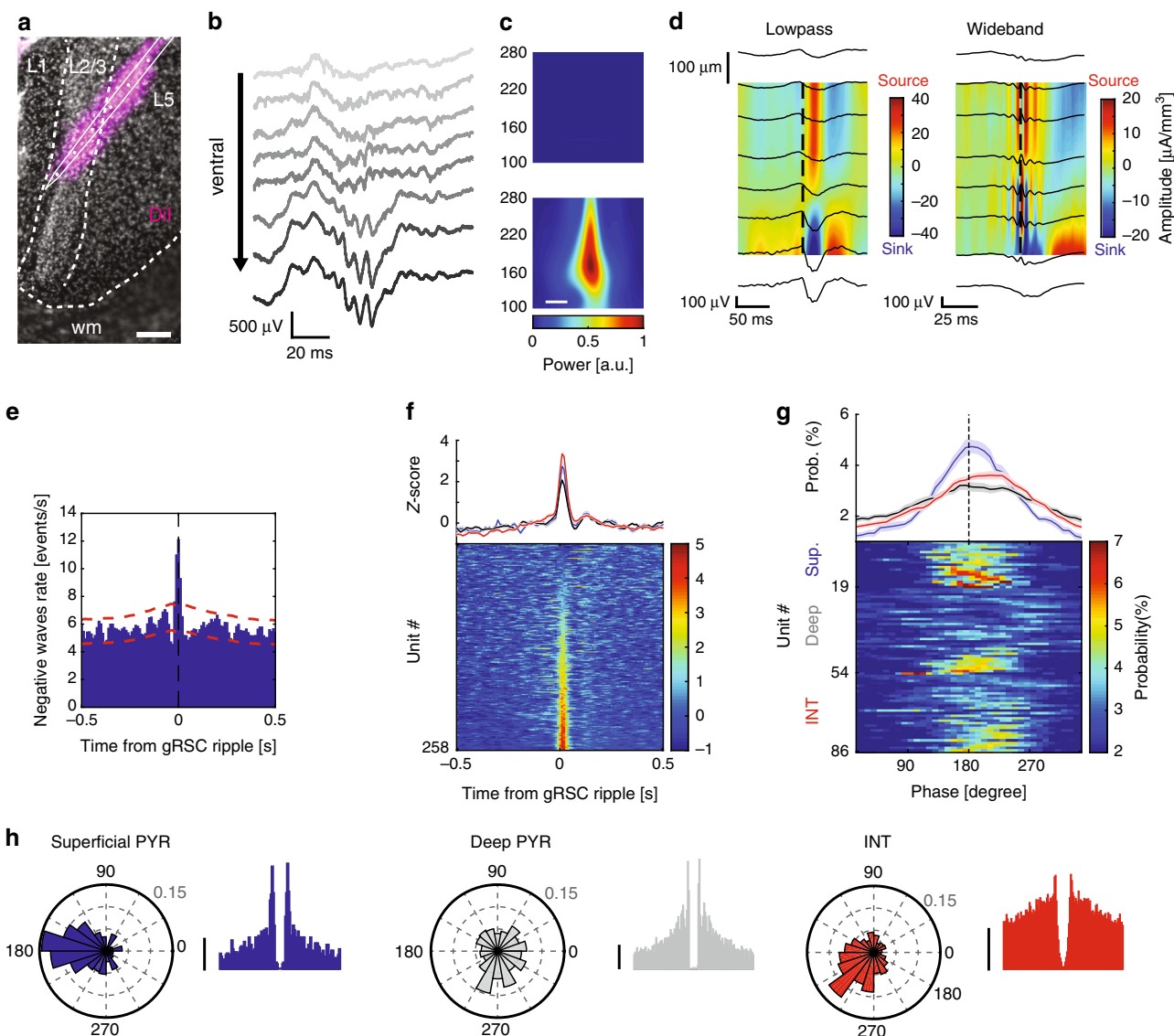

**Fig. 1 Expression of local ripple activity in the gRSC. a** Example histological verification of probe location. wm, white matter. Scale bar, 100 μm. **b** Laminar profile from eight channels ranging from dorsal (top) to ventral (bottom), showing a stronger expression of ripple activity in superficial layers. **c** Ripple power is increased in superficial (bottom), compared to deep layers (top). Wavelet spectrograms centered around ripple peak averaged across 312 events from one example recording. Scale bar, 10 ms. **d** CSD analysis from two example sessions showing low-passed filtered (left) and broad-band (right) versions centered around gRSC ripple peak (dashed line; 496 and 288 events, respectively). **e** Average cross-correlogram between cortical ripple events and negative waves in superficial gRSC ($n = 8$ sessions from four animals; 1914 ripples, 72,422 negative waves). Red lines, 95% confidence intervals. **f** Z-score normalized PETH showing the average (mean ± s.e.m; top) and individual (bottom) responses of 258 gRSC units ($n = 19$ sessions from six mice) to cortical ripples sorted into superficial pyramids (blue), deep pyramids (black) and interneurons (red). **g** Average (top; mean ± s.e.m) and color-coded histograms (bottom) of preferred gRSC ripple phase of putative superficial pyramids (blue, top rows), deep pyramids (gray, middle rows) and INs (red, bottom rows) that are significantly phase-modulated based on circular Rayleigh test (superficial pyramids 19/51 = 37%; deep pyramids 35/174 = 19%; INs 32/110 = 29%). **h** Example polar plots of phase-locking to gRSC ripples (left) and the corresponding ACGs (right) of a putative superficial pyramid (left panel blue), deep pyramid (middle panel gray) and IN (right panel red). Radial axis: probability. ACGs range ±25 ms, scale bar 5 Hz.

complement the CSD analysis, the sink-source profile was validated using independent component analysis (ICA) for blind source separation, which identified an increase in voltage load in superficial layers. To quantify the relation between CA1 SPW-Rs and gRSC ripples, we calculated their wavelet phase- and power coherence. Both measures showed a peak in the ripple frequency band (Fig. 2j and Supplementary Fig. 3), suggesting a fine-timescale coupling between these high-frequency oscillations.

Unit firing in gRSC was also time-locked to hippocampal SPW-R events, with responses of individual units showing either a single peak or drop in firing rate, or more complex SPW-R-

related discharge patterns (Fig. 3a–c). At the population level, SPW-R-associated responses were characterized by a peak locked to SPW-R onset time, and were often both preceded and followed by secondary peaks at approximately 100 ms from SPW-R onset. These secondary peaks were absent from the peri-event time histograms (PETHs) of CA1 units (Supplementary Fig. 3), suggesting that this gRSC activity was unlikely to have been driven by the hippocampus. Compared to deep gRSC units, superficial neurons responded earlier after SPW-R (mean latency to peak of superficial neurons: 23 ± 2 ms; deep: 32 ± 3 ms, Fig. 3a, b) and showed stronger SPW-R event modulation (37/51 of

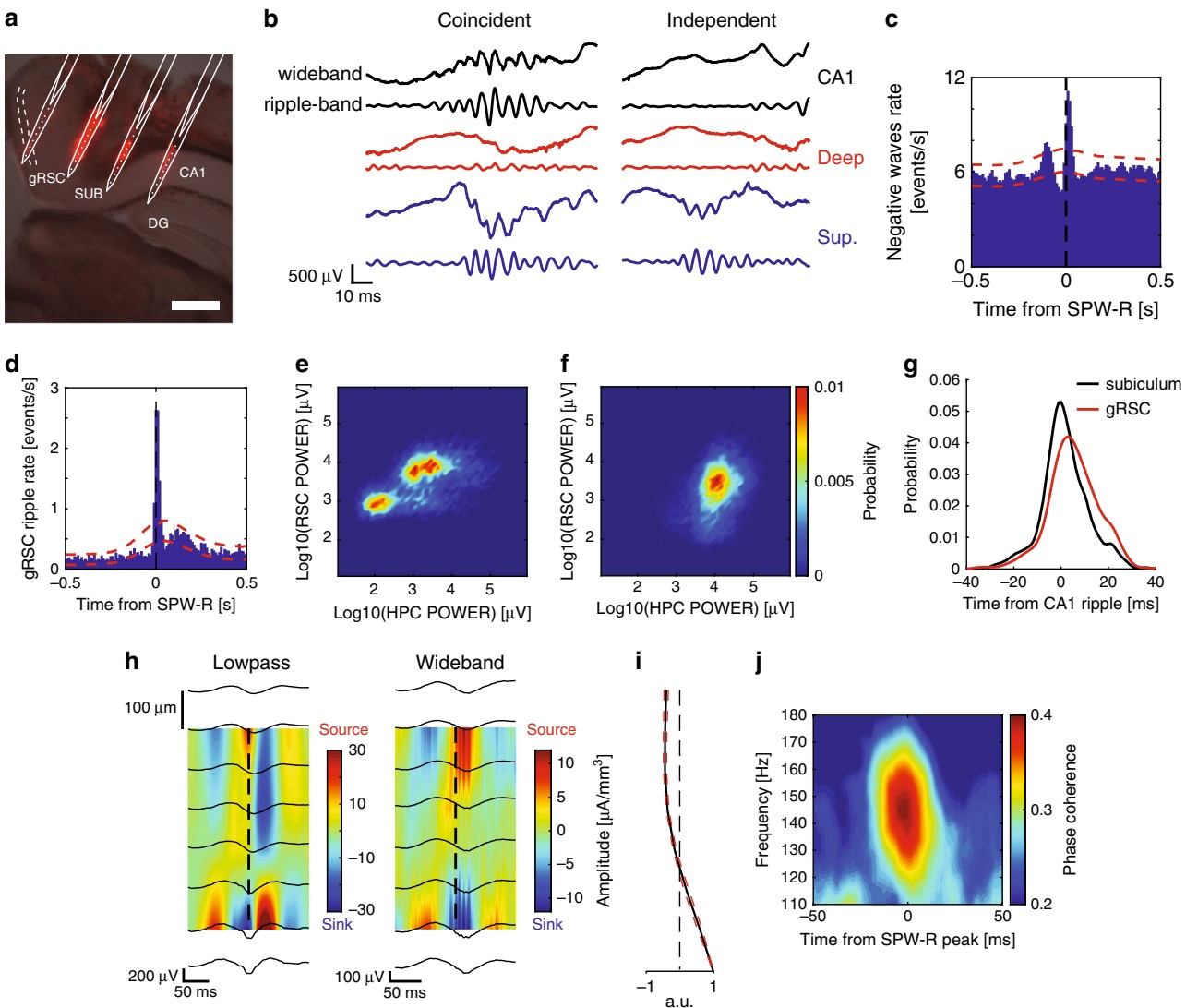

**Fig. 2 gRSC activity is tightly coupled to hippocampal SPW-Rs. a** Example histological verification of probe location (DiI, red). Relevant areas are labeled; L2/3 is marked by dashed lines. Scale bar, 500 µm. **b** Two examples of gRSC ripples showing traces of wide-band (top trace) and ripple band (bottom trace) filtered activity from CA1 area (top, black), deep layers gRSC (middle, red) and superficial layers gRSC (bottom, blue). **c** Average cross-correlogram between SPW-R events and negative waves in superficial gRSC ($n = 8$ sessions from four animals; 3409 SPW-Rs, 72,422 negative waves). Red lines, 95% confidence intervals. **d** Average cross-correlogram between CA1 (reference) and gRSC ripples ($n = 18$ sessions from six animals; 22,694 CA1 SPW-Rs, 8501 gRSC ripples). **e** Density distribution of peak ripple band power correlations between CA1 and superficial gRSC recording sites for gRSC ripples (least-square slope = 0.40; $\rho = 0.60$; $p < 10^{-10}$, two-sided Student's $t$-test). **f** Same as (**e**) but for CA1 ripples (least-square slope = 0.42; $\rho = 0.31$; $p < 10^{-10}$). **g** Probability distribution of subicular and gRSC ripple peak power lags relative to CA1 peak power ($n = 4$ sessions from two mice where simultaneous targeting of CA1, subiculum and superficial gRSC was histologically confirmed; median lag of subicular ripples: $0.80 \pm 0.32$ ms; gRSC ripples: $4.76 \pm 0.34$ ms; $p = 1.5 \times 10^{-17}$, two-sided signed rank test). **h** CSD analysis from two example sessions showing low-passed filtered (left) and broad-band (right) versions centered around CA1 SPW-R peak (dashed line; 821 and 430 events, respectively). **i** ICA (mean ± s.e.m) decomposition of voltage traces from the gRSC centered around CA1 SPW-R showing an increase in voltage loadings in superficial layers ($n = 8$ sessions from seven animals; $p = 0.000155$, two-sided rank sum test between first IC loads of top and bottom channels). **j** Wavelet phase coherogram between CA1 and superficial gRSC LFP centered around CA1 SPW-R detection (7150 events from seven animals).

superficial neurons, 72% were significantly modulated compared to 51/174, 29%, deep neurons). A large fraction of putative gRSC interneurons were also significantly affected (Fig. 3c; 78/110, 71%). Interneurons exhibited heterogeneous responses suggesting a cell type-specific modulation by SPW-Rs similar to hippocampal interneurons[37]. When individual cycles of hippocampal SPW-Rs were used as a reference, significant phase-locking was observed in a considerable fraction of superficial pyramids and interneurons, but not deep pyramids (Fig. 3d, e; superficial 15/51, 29%; deep 14/174, 8%; interneurons 21/110, 19%). These results were supported by a spike-LFP coherence analysis, which

indicated an increase in ripple band coherence in superficial, but not in deep layers (Fig. 3f).

To further test the influence of SPW-R activity on hippocampal—gRSC communication, we examined the co-modulation of CA1 gRSC unit pairs during and outside of SPW-R epochs by computing their standardized cross-covariance[38]. Peak cross-covariance was higher within SPW-R epochs (250 ms before and after ripples with CA1 units as reference) compared to periods outside SPW-Rs for both superficial and deep units, and peaked at positive values, suggesting an increase of excitatory drive from the CA1 to the gRSC at times of SPW-Rs. In line with the

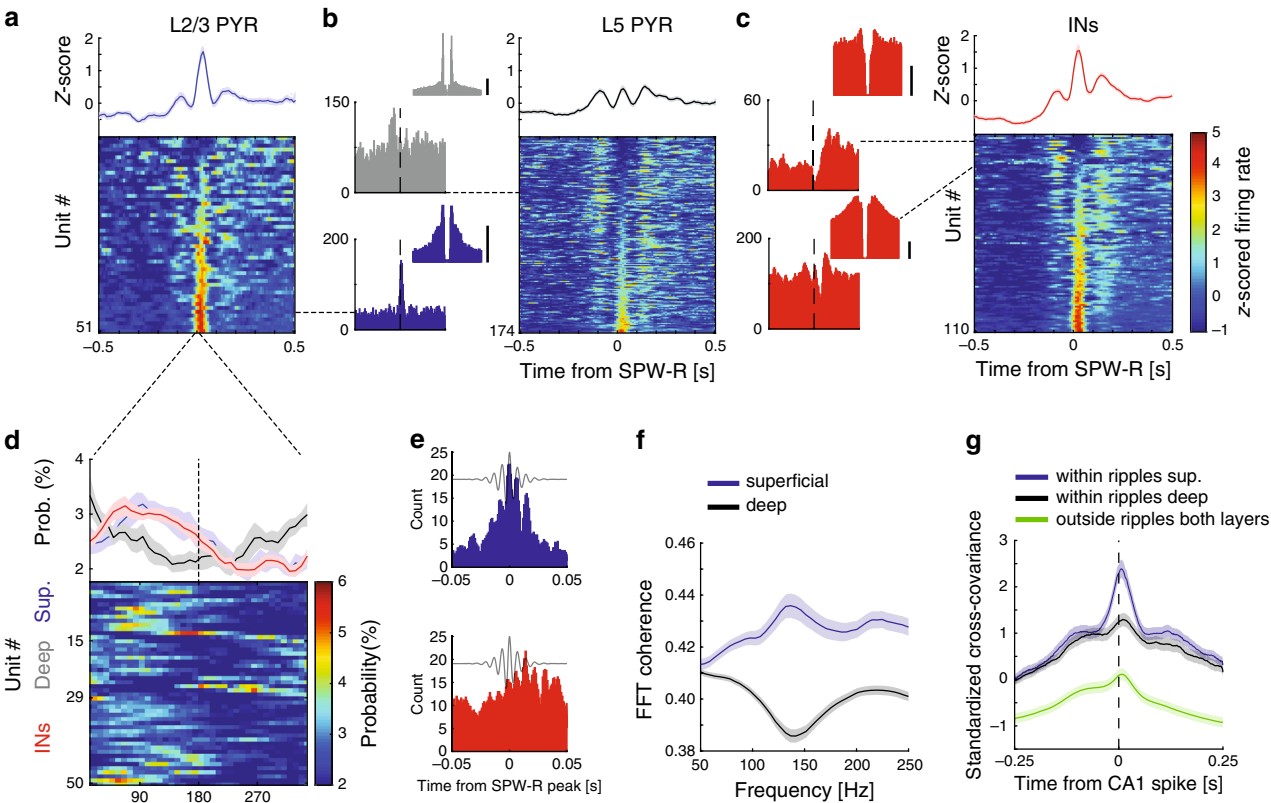

**Fig. 3 Modulation of gRSC units by hippocampal SPW-Rs is layer and cell-type dependent. a–c** Z-score normalized PETHs showing the average (mean ± s.e.m; top) and individual (bottom) responses of superficial pyramidal neurons (**a**), deep pyramidal neurons (**b**) and interneurons (**c**) sorted based on ripple modulation ($n = 27$ sessions from eight mice). Dashed lines point one example PETH and the corresponding ACG (±25 ms). Scale bars, 10 Hz; Y-axis: spike count. **d** Average (top) and color-coded histograms (bottom) of preferred CA1 ripple phase of superficial pyramids (blue, top rows), deep pyramids (gray, middle rows) and INs (red, bottom rows) that are significantly phase-modulated based on circular Rayleigh test (superficial pyramids 15/51 = 29%; deep pyramids 14/174 = 8%; INs 21/110 = 19%). **e** High resolution CA1 SPW-R triggered PETH of an example superficial unit (top, blue) and an IN (bottom, red) significantly modulated by individual CA1 ripple waves. **f** Average spike-LFP coherence between superficial (blue) and deep (black) units and CA1 LFP (±100 ms around SPW-R peak) showing a strong tuning of superficial, but not deep, cells to the ripple band ($n = 27$ sessions from eight animals). **g** Standardized cross-covariance during SPW-Rs averaged over pairs of superficial (black) and deep (red) gRSC units firing within and outside of SPW-Rs epochs (green, averaged across both layers) ($n = 27$ sessions from eight animals; 209 CA1–superficial gRSC pairs, 351 CA1–deep gRSC pairs and 560 CA1–gRSC units outside ripples). See Supplementary Fig. 3.

SPW-R-induced responses (Fig. 3a, b), CA1 spiking co-varied stronger with superficial gRSC unit activity than with activity in deep gRSC. Superficial gRSC also exhibited a shorter latency to peak covariance (8.5 ms ± 0.4 ms), compared to spiking activity of units in deeper layers (12 ± 0.3 ms; Fig. 3g and Supplementary Fig. 3). In summary, these data suggest that modulation of gRSC neurons by hippocampal SPW-Rs is layer-specific and that a substantial amount of information is transferred between these structures during SPW-Rs.

**Functional topography between the hippocampus and gRSC.** We next explored the route taken by CA1 SPW-R to reach gRSC targets. To this end, we performed parallel recordings using 8-shank 256-site silicon probes positioned along the anterior-posterior axis of the gRSC and either 8-shank 256-site (mounted with optic fibers) or 4-shank 32-site (with integrated μLEDs; see Methods) silicon probes positioned along the subiculo-fimbrial axis of the dorsal hippocampus in awake head-fixed CaMKII-Cre:: Ai32 mice ($n = 3$; Supplementary Fig. 5). We induced artificial local oscillations optogenetically at specific anatomical locations in dorsal CA1 and subiculum (Fig. 4a). Optogenetic activation (100 ms square pulse) of CA1 evoked strong oscillatory activity (Supplementary Fig. 5), termed as induced high-frequency oscillations (iHFOs)[39]. For these mapping experiments we used

relatively strong light intensity that induced large amplitude but relatively low frequency oscillation (possibly due to cycle skipping and ChR2 deactivation; Supplementary Fig. 5). iHFOs could be induced at any given location along the subiculo-fimbrial axis of the CA1 str. pyramidale with a similar magnitude (Supplementary Fig. 5). Stimulation in the subiculum or at the subicular end of CA1 (CA1a) reliably evoked a series of concatenated large-amplitude negative waves in the gRSC, which increased in amplitude with increasing stimulation intensity (Fig. 4b left and Supplementary Fig. 5). In contrast, stimulation of more proximal sites of CA1 induced smaller or no discernible gRSC responses (Fig. 4b, right and Supplementary Fig. 5). Hippocampal LFP responses recorded from shanks away from the site of stimulation decayed to baseline levels within ~400 μm, ruling out the possibility of directly activating the gRSC (Supplementary Fig. 5). To quantify the magnitude of evoked responses for each stimulation site, evoked gRSC potentials were normalized with respect to the locally induced CA1 responses, which indicated a gradual proximal-distal increase in response magnitude (Fig. 4c). The magnitude of evoked responses in the gRSC depended not only on the site of CA1 stimulation but also on the distance from the stimulation site, with more anterior portions of the gRSC (bordering the anterior cingulate cortex, ACC) exhibiting weaker responses (Supplementary Fig. 5). Similar to the responses to

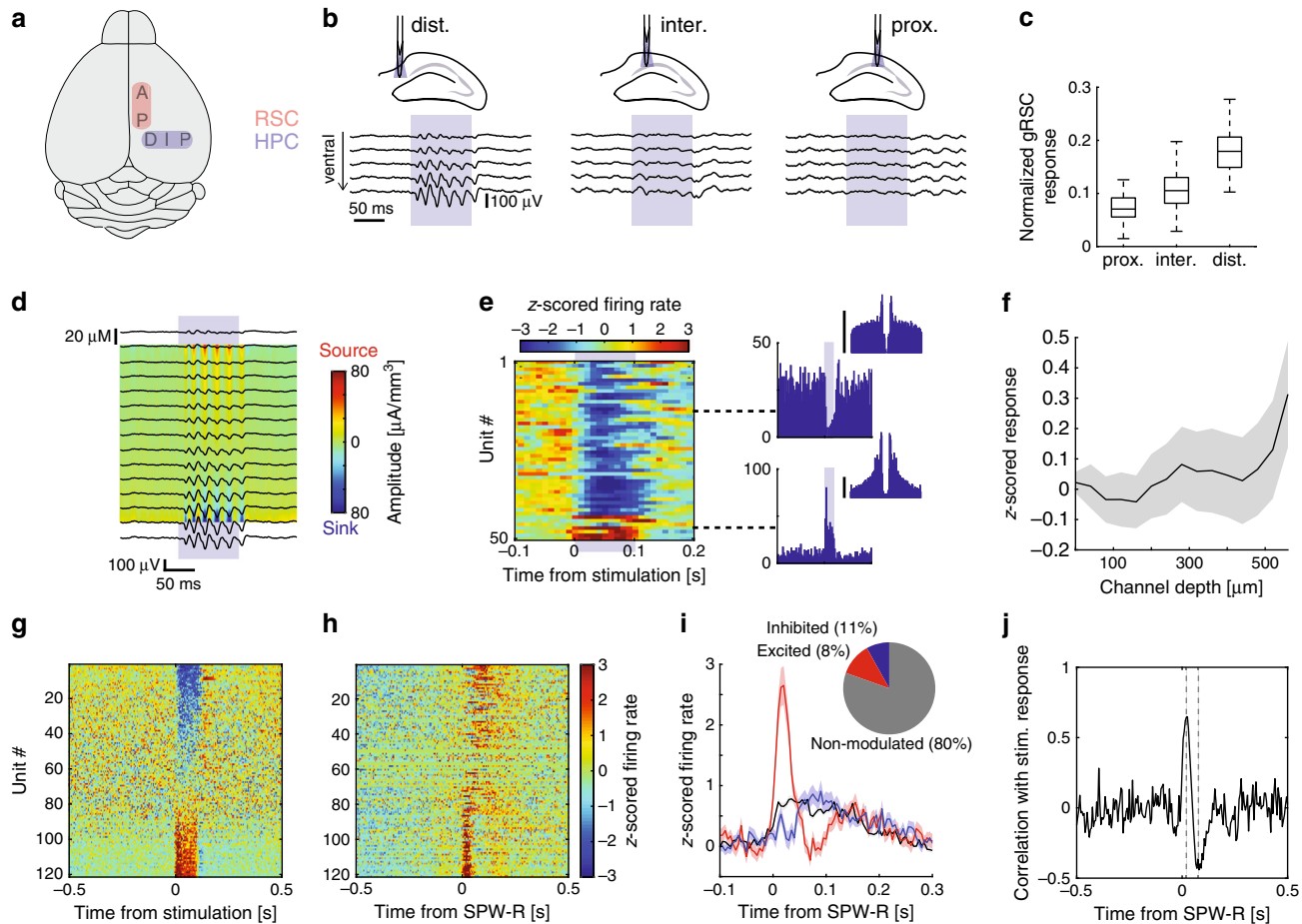

**Fig. 4 Mapping of functional topography between dorsal hippocampus and gRSC in CaMKII-Cre::Ai32 mice. a** Schematic depiction of probe placement. Two 8-shank probes were used to cover 1050 μm of dorsal CA1 and subiculum medial-lateral axis and the gRSC anterior-posterior axis. **b** Voltage traces (mean ± s.e.m) from five channels ranging from dorsal (top trace) to ventral (bottom trace) showing gRSC LFP responses to distal (left), intermediate (middle) and proximal (right) CA1 stimulation averaged over >300 events from one animal. **c** Normalized gRSC response to stimulation expressed as the ratio between local responses in CA1 and subiculum and evoked responses in the gRSC from one example animal ($n > 100$ stimulations), plotted as a function of the location of stimulation along the proximal-distal axis of the dorsal hippocampus. Data are displayed as box plot representing median, lower and upper quartiles and whiskers representing most extreme data points. **d** CSD analysis of responses to distal CA1 stimulation and the corresponding voltage traces averaged over >300 events from one animal. **e** Left: Normalized raster plot of gRSC unit firing in response to distal CA1 stimulation from one shank, sorted based on the location of maximal waveform amplitude. Right: individual examples of a deep inhibited pyramid (top) and a superficial excited pyramid (bottom). *Y*-axis, spike count; scale bar, 5 Hz. **f** Summary of *z*-scored firing responses to distal CA1 stimulation as a function of maximal waveform depth (mean ± s.e.m., $n = 3$ sessions from three animals). **g** Z-score normalized responses of gRSC units significantly modulated by CA1 stimulation sorted based on response magnitude. **h** Z-score normalized responses of gRSC units to CA1 SPW-Rs sorted as in (**g**). **i** Averaged *z*-scored firing rate in response to stimulation of gRSC units which are positively (red), negatively (blue) or not modulated (black, not shown in (**g**) and (**h**)) by hippocampal stimulation (mean ± s.e.m; $n = 619$ units from three mice and three sessions). Inset, proportions of each group. **j** Firing time from SPW-R onset is correlated with response magnitude. Dashed gray lines mark maximal and minimal correlation.

spontaneous SPW-Rs, induced responses were also more prominent in superficial layers, as indicated by the strong sinks in the CSD map (Fig. 3d). We recorded 619 gRSC units during optogenetic hippocampal stimulation. The majority of excited units were located at superficial recording sites, whereas firing rates of many deep layer units were suppressed (Fig. 4e, f). Stimulation-excited units ($n = 50/619$, 8%) were recruited earlier after SPW-R onset, compared to inhibited units ($n = 71/619$, 11%; Fig. 4g–i). The degree of early firing during SPW-Rs correlated with the degree of stimulation-induced firing, while late firing was correlated with stimulus-induced inhibition (Fig. 4j). These results indicate a functional topography between CA1, the subiculum and gRSC. They also suggest that, while gRSC responses to hippocampal activity are governed by direct excitation in superficial layers, feedforward inhibition dominates in deep layers, which

together influence the sequential response of gRSC neurons to SPW-Rs.

**A subset of gRSC neurons are preferentially recruited by unique hippocampal activity patterns.** A key question is whether gRSC neurons respond specifically to unique combinations of hippocampal activity during SPW-Rs, or whether SPW-Rs provide the necessary level of excitation needed for the induction of local ripples that, in turn, self-organizes neuronal content[7,40–42]. To address this question, we used a k-means ($k = 10$; Supplementary Fig. 6) clustering algorithm to classify CA1 SPW-Rs in our high-density recordings ($n = 3$ sessions from three mice, 5442 ripples) based on the population vector of firing rates of CA1 neurons during the SPW-R. The resulting similarity matrix

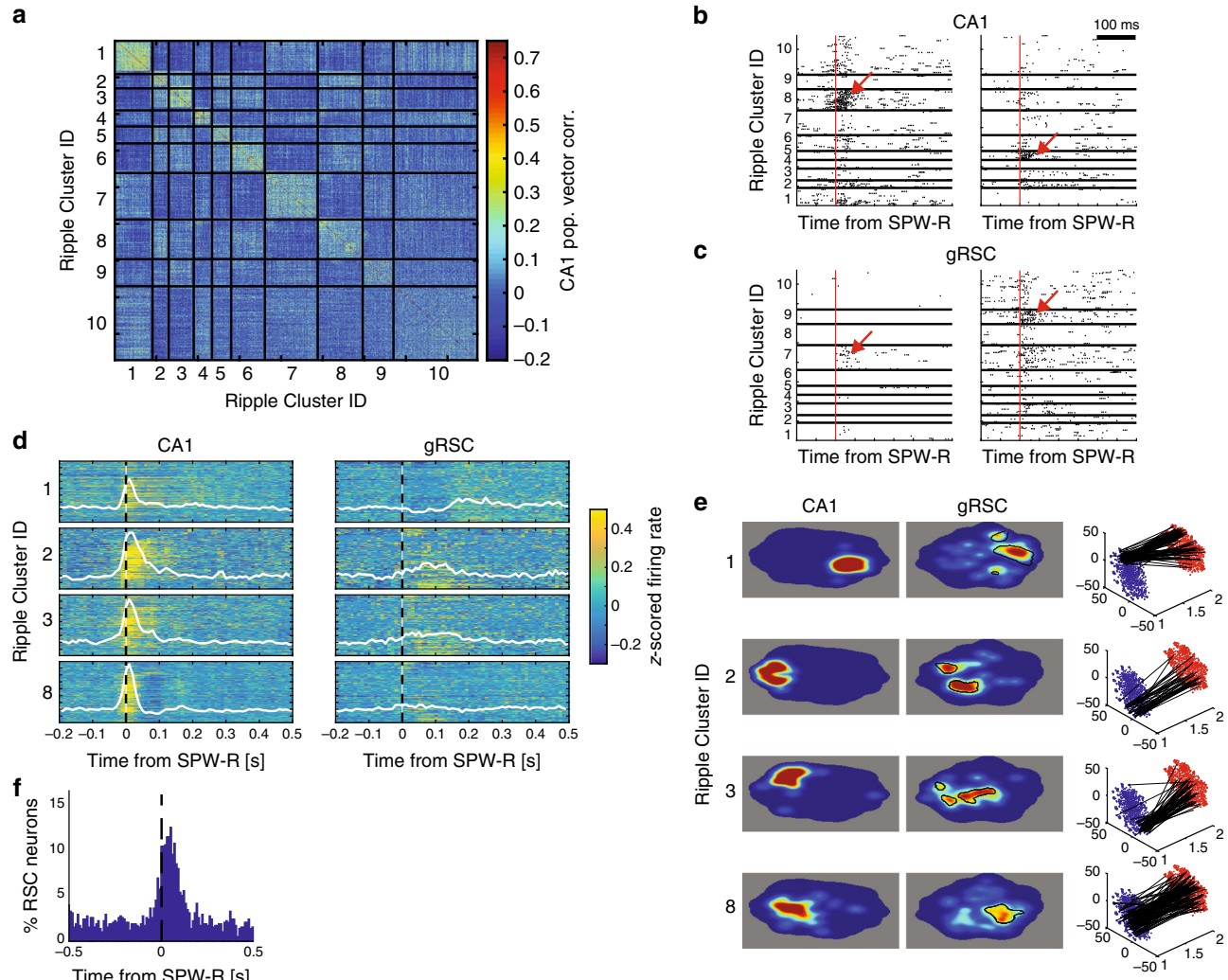

**Fig. 5 Retrosplenial activity distinguishes CA1 ripple firing patterns. a** Example similarity matrix showing the correlation of population firing rate vectors of CA1 ensembles during ripples. Matrix sorted by ripple cluster labels ($N = 871$ ripples). **b** Two example CA1 neuron responses to different ripple types (red arrows). Red line, ripple onset. **c** As in (**b**) for two example gRSC neurons. **d** Population responses to four classes of ripple. Each row of each panel shows the normalized firing rate of an individual neuron around ripple onset, averaged across all ripples of the same type. White line, population average. **e** Two-dimensional histograms of the t-SNE representation for each ripple cluster type shown in **d** for CA1 (left), gRSC (middle), and their event-by-event correspondence (right, black lines). Black contours in the gRSC denote the region of t-SNE space with higher density than expected by chance ($p < 0.01$, permutation test with 1000 shuffles; see Methods). See Supplementary Fig. 6 for additional examples. **f** Percentage of retrosplenial neurons that discriminate ripple types at each time lag around ripple onset.

(Fig. 5a) indicated varying degrees of differences of the clustered SPW-R categories, likely reflecting different subsets as well as different weightings of active neurons. SPW-Rs of a given cluster were distributed throughout the entire recording session in all animals (Supplementary Fig. 6). By definition, the activity of CA1 neurons strongly discriminated different SPW-R subtypes, responding to only one or few ripple clusters (Fig. 5b, d). Importantly, we found that the activity of gRSC neurons ($n = 619$ units) also discriminated CA1 SPW-R types. Individual neurons responded maximally to specific SPW-Rs types or the combination of their subsets (Fig. 5c, d). As a result, each SPW-R type was associated with specific population firing patterns of gRSC neurons, with some gRSC neurons showing opposing activity patterns with respect to different clusters of SPW-Rs (Fig. 5d). Those differences were also preserved in the low-dimensional representation of the different ripple clusters and their associated gRSC responses (Fig. 5e). Overall, 46.8% of gRSC neurons discriminated SPW-R type at some point within the 100 ms after ripple onset (Fig. 5f; $p < .01$; ANCOVA with 10 ripples groups

and CA1 population firing rate as a continuous nuisance regressor), with the majority of neurons discriminating SPW-R type within 60 ms after SPW-R onset. These findings were validated using non-parametric shuffling analysis (Supplementary Fig. 6).

**SPW-R to gRSC communication depends on brain state**. The probability of occurrence of hippocampal SPW-Rs is biased by ongoing brain activity, such as UP and DOWN states of slow oscillations of sleep[43,44] and sleep spindles[8,21,36,45,46]. However, the influence of brain state changes on neurotransmission in the immobile awake animal is less clear.

Unlike hippocampal LFP patterns, which classify overt behaviors into preparatory ("voluntary") and consummatory behaviors, including immobility, by shifting from theta to non-theta patterns[47], neocortical LFP recordings typically show "desynchronized" LFP during both types of waking behaviors[48]. However, during waking immobility, an additional prominent

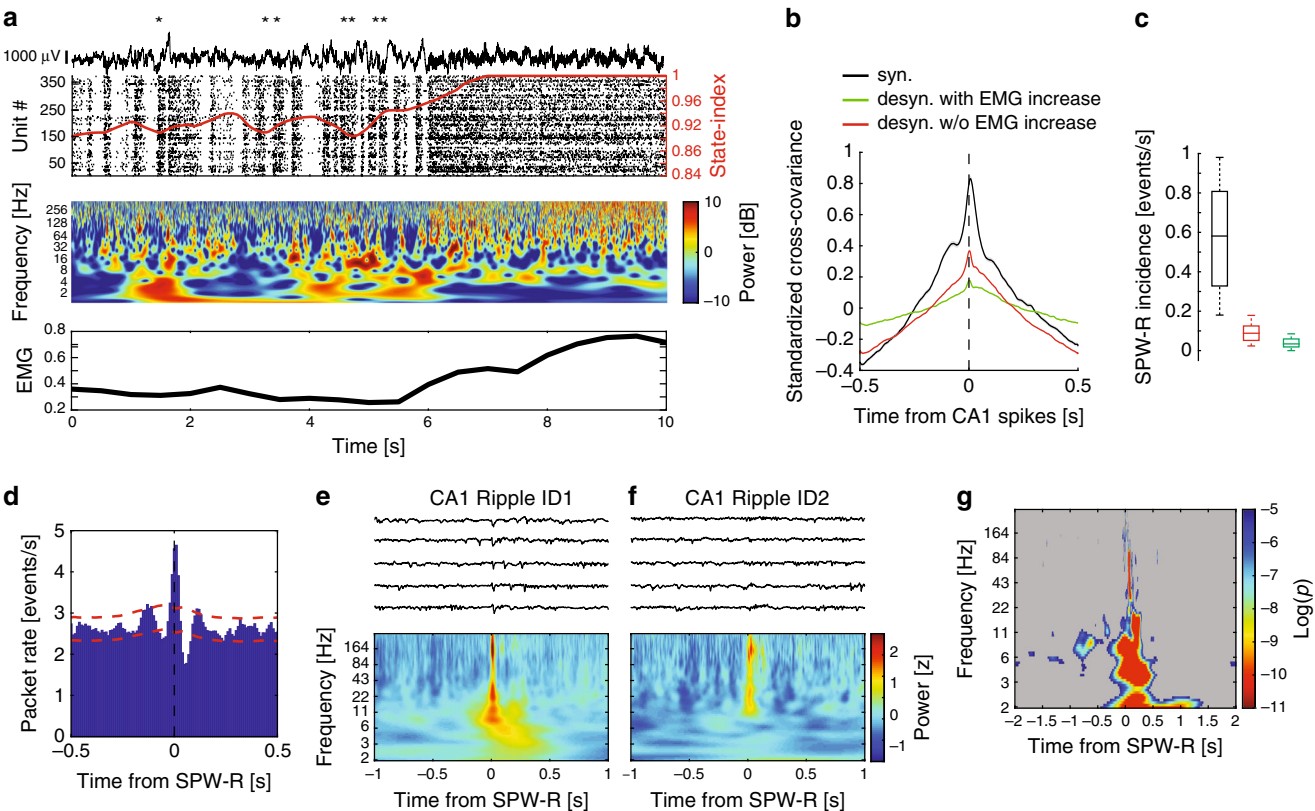

**Fig. 6 Hippocampal-gRSC communication is state dependent. a** Top panel: example of 10 s gRSC LFP (top), spiking activity (bottom) and the corresponding state-index (red) showing state transition from a synchronized to a desynchronized state. Asterisks, SPW-Rs detected in CA1. Middle panel: Wavelet spectrogram of the LFP trace shown above. Note the absence of low frequency activity in the desynchronized state and the alternating high-frequency activity in the synchronized state reflecting population bursts. Bottom panel: corresponding EMG activity. See also Supplementary Fig. 7. **b** Standardized cross-covariance between dorsal CA1 and subiculum and gRSC unit pairs taken from synchronized (black) and desynchronized regimes either with (green) or without (red) increase in EMG activity ($n = 12$ sessions from five animals and 19,064 unit pairs). See Supplementary Fig. 7. **c** SPW-R rate is reduced during desynchronized compared to synchronized regimes. Data are displayed as box plot representing median, lower and upper quartiles and whiskers representing most extreme data points ($n = 10$ sessions from four animals). **d** Cross-correlogram between hippocampal SPW-Rs and gRSC activity packets ($n = 14$ sessions from three CaMKII::Ai32 and four VGlut2-Cre mice, 8924 SPW-Rs and 87,820 packets). **e** Top: Example LFP traces from most superficial and posterior gRSC electrode time locked to the onset of CA1 ripple cluster ID1. Bottom: Mean spectrogram for cluster ID1. **f** Same as (**e**) for ripple cluster ID2. **g** The degree to which the power of each frequency band at each lag around ripple onset distinguishes ripple cluster ID. Non-significant areas are shown in gray. Corrected for multiple comparisons at each time ($p < 0.0004$, ANCOVA test with Bonferroni correction).

population pattern, characterized by ~6–10 Hz LFP oscillation, occasionally alternates with desynchronized cortical LFP[49]. At the neuronal level the synchronized pattern is characterized by alternating population firing and silent periods, previously referred to as activity packets[50,51] (Fig. 6a). During the silent periods, the activity of most cortical neurons is strongly reduced, reminiscent of the DOWN state of NREM sleep (Supplementary Fig. 7b)[50–54]. We defined activity packets as events where bursts of population activity were surrounded by near silence and reached at least 60% of its long-term average for more than 40 ms (Methods). gRSC population activity fluctuated between synchronized epochs marked by 3–10 Hz activity packets and a desynchronized state associated with high or low electromyographic activity (segregated using cutoffs at the minimum of the bimodal distribution[55]) (Fig. 6a and Supplementary Fig. 7). These fluctuations were quantified using the state-index measure[51] (Methods). Synchronized epochs were associated with low frequency activity reflecting negative waves as well as bursts of gamma-band and high-frequency activity, reflecting population bursts (Fig. 6a and Supplementary Fig. 7a, c, d).

We next asked how ongoing cortical activity affects the efficacy of hippocampal-cortical coupling during SPW-Rs. CA1-gRSC population cross-covariance was larger during synchronized states (Fig. 6b), compared to both desynchronized epochs with or without increase in electromyographic activity ($p < 0.001$), despite similar overall firing rates in each state in both structures (Supplementary Fig. 7). SPW-Rs were virtually absent during desynchronized states associated with increased EMG and theta oscillation (Fig. 6c and Supplementary Fig. 7)[35,54]. Analogous to the coupling of SPW-Rs to NREM UP and DOWN states[19,36,45], hippocampal SPW-Rs were significantly coupled to the onset of cortical packets in the waking state (Fig. 6d).

Brain state changes could also contribute to the differential readout of SPW-Rs by gRSC neurons. To examine this possibility, we tested whether wavelet coefficients at each time point surrounding hippocampal SPW-R onset differentiated SPW-R type (Fig. 5), while regressing out the effect of CA1 population firing rate (Supplementary Fig. 7j, Methods). After SPW-R onset, divergent spectral LFP patterns were observed according to SPW-R subtype ($n = 3$ mice, one session per mouse), with significant differences in low frequency bands, reflecting the occurrence of a cortical wave (compare Fig. 6e, f; see also Supplementary Fig. 7i) and in higher bands reflecting gamma band power and spiking. In contrast to the unit discrimination of ripple type, which proceeded ripple onset (Fig. 5e), LFP magnitude in the 6–10 Hz frequency range prior to SPW-R onset significantly differed

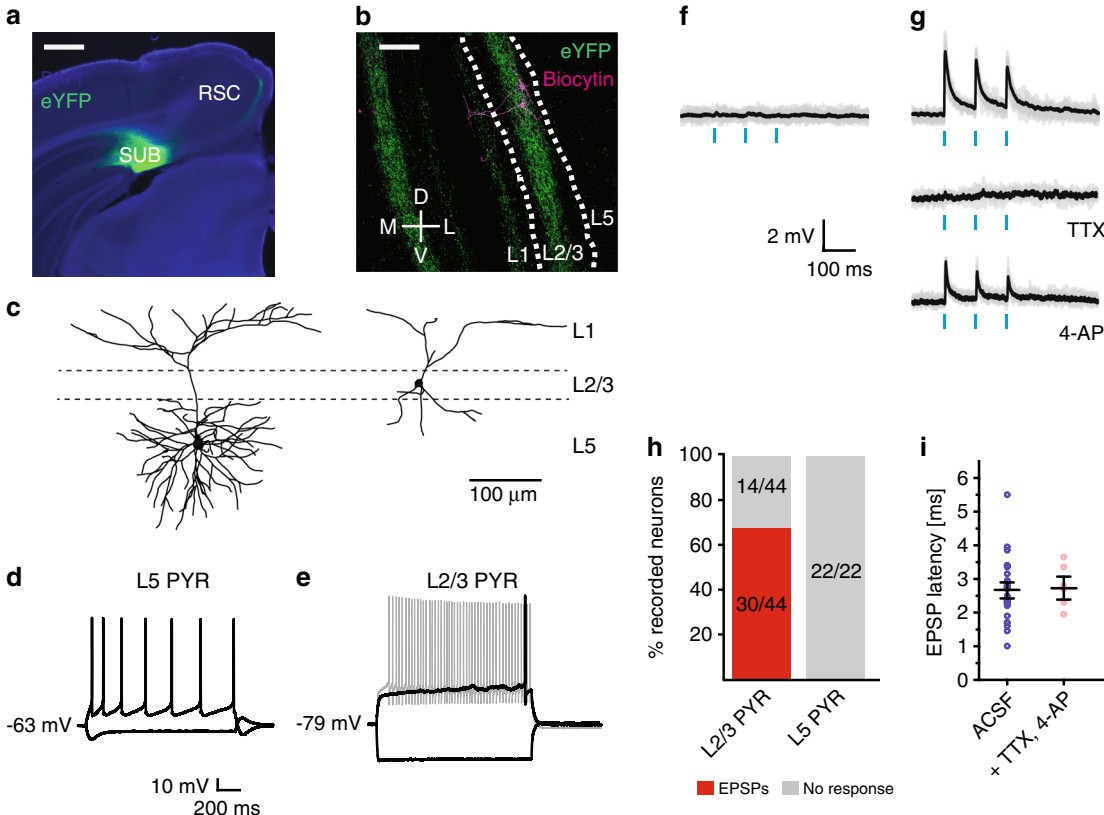

**Fig. 7 Connectivity between subicular VGlut2+ bursting cells and the gRSC, characterized in vitro. a** Infection of VGlut2+ subicular cells with AAV-FLEX-ChR2-YFP in VGlut2-Cre mice reveals strong projections onto superficial layers of the gRSC (green, eYFP; blue, DAPI). Scale bar, 500 μm. **b** Magnified image showing the (bilateral) restriction of subicular VGlut2+ terminals predominantly to L2/3 (green), as well as two biocytin-labeled L2/3 pyramids (magenta). White lines, L2/3. Scale bar, 100 μm. **c** Reconstructed representative examples of deep (left) and superficial (right) gRSC pyramids. Superficial cell is the one shown in (**b**). **d** Membrane potential responses of a deep gRSC pyramid to injection of ±80 pA. **e** Responses of a superficial cell to current injection of ±80 pA (black) and 160 pA (gray). **f** Example traces of responses to light stimulation of a deep gRSC pyramid in ACSF. **g** Example responses of a L2/3 gRSC pyramidal cell in ASCF (top), after adding 10 μM TTX (middle) and after the addition of 4-AP (bottom). **h** Summary of responses of gRSC pyramids to subicular VGlut2+ fiber light stimulation. i) EPSP latencies (mean ± s.e.m) of all responsive superficial cells (n = 30) and a subset where TTX and 4-AP were applied (n = 6), indicating monosynaptic connections.

according to ripple subtype (Fig. 6g). These results suggest that the cortical state at the time of SPW-R influences which hippocampal ensembles are active, which in turn affects which gRSC neurons fire in response to hippocampal drive.

**Subicular bursting neurons target gRSC**. The hippocampus-gRSC functional route likely involves the subiculum, since bursting cells at the distal subiculum[56] give rise to projection to the gRSC[30,57,58]. To examine the involvement of this route, we used the VGlut2-Cre mouse line, which expresses Cre in subicular bursting cells[59], and shows a proximo-distal expression gradient in the dorsal subiculum[60]. We first employed a ChR2-assisted circuit mapping strategy, by injecting AAV9-SwitchON-mRu-byNLS-ChR2(H134R)-EYFP in the dorsal subiculum of VGlut2-Cre mice. The design of this switch vector allows for the expression of ChR2-YFP in Cre-positive cells and mRuby in Cre-negative cells, and was useful in ruling out viral spread to the gRSC, where VGlut2 is also sparsely expressed (Allen Brain Atlas, experiment #73818754) (Supplementary Fig. 8). In all mice used for this experiment (n = 8), viral expression was limited to the dorsal subiculum and showed dense axonal termination in L2/3 gRSC and strongly co-localized with VGlut2-reactive fibers (Fig. 7a, b and Supplementary Fig. 8). VGlut2+ axons extended across the entire anterior-posterior axis of the gRSC but exhibited a gradient in intensity, reaching their maximum above the

splenium and decreasing toward more rostral parts at the border with the ACC, explaining the gradually weakening responses along this axis to hippocampal stimulation. VGlut2+ fibers sharply terminated at the border with the ACC and at the border between the granular and dysgranular RSC (Supplementary Fig. 8).

In cortical slices in vitro, we recorded from 73 cells intracellularly, which were subdivided into L2/3 pyramids (n = 44), L5 pyramids (n = 22) and fast spiking interneurons (n = 7, five in L2/3 and two in L5) based on morphology, firing pattern and, in a subset of the experiments, GAD-67 immunostaining (Supplementary Fig. 9). L2/3 pyramids were smaller than L5 pyramids, and exhibited a late-spiking phenotype, a strongly hyperpolarized resting membrane potential, and no sag potentials (Fig. 7c–e; Supplementary Table 1)[61]. Brief, 10 ms pulses of blue light evoked excitatory postsynaptic potentials (EPSPs) in 68% (30/44) of L2/3 cells (Fig. 7g–h), with a mean latency of 2.66 ± 0.15 ms (Fig. 7i). In a subset of the experiments, we blocked action potential evoked neurotransmission by tetrodotoxin to confirm that these connections were monosynaptic. After all light-evoked responses were abolished, we added the K+-channel blocker 4-AP to the extracellular medium to rescue monosynaptic transmission (Fig. 7g). Monosynaptic coupling between subicular VGlut2+ fibers and gRSC was confirmed for all tested L2/3 pyramidal neurons (n = 6/6). Likewise, light stimulation of subicular fibers in superficial layers reliably evoked EPSPs in all

tested interneurons ($n = 5/5$, Supplementary Fig. 9). In stark contrast to superficial layers, and in contrast to a previous report[62] (likely arising from the minimal injection volume and the ability to assess viral spread using the Switch vector in our study) YFP-labeled subicular fibers in deep layers were extremely sparse and light pulses delivered either to somatic or apical regions failed to evoke measurable responses in deep pyramidal cells (Fig. 7f, h). In contrast to deep layer pyramids, we did observe light-evoked responses in deep layer INs ($n = 2/2$; Supplementary Fig. 9). To validate that subicular neurons that project to gRSC are indeed mainly bursting cells[56,63], we injected a non Cre-dependent rAAV2-retro-tdTomato virus[64], a recently developed retrograde monosynaptic tracer, into the gRSC of one VGlut2-Cre mouse. Labeled neurons were restricted to the dorsal subiculum and showed a bursting phenotype ($n = 8$ cells; Supplementary Fig. 9).

Reciprocally, we tested whether VGlut2-negative, putative non-bursting cells give rise to similar projection pattern in the gRSC downstream responses. We injected Cre-Off ChR2-EYFP in the dorsal subiculum of VGlut2-Cre mice ($n = 3$) and measured light-evoked synaptic responses in superficial and deep gRSC cells. When holding cells at a more depolarized membrane potential (-50 mV), we found that light stimulation evoked mostly inhibitory responses in both superficial (15/24, 62%) and deep cells (7/11, 63%) (Supplementary Fig. 9). In contrast, EPSPs were evoked in only a small subset of superficial (4/24, 16%) and deep (2/11, 18%) cells, with the remaining neurons showing no light-evoked responses. Furthermore, subicular terminals labeled by the Cre-off vector (i.e. stemming from non-VGlut2$^+$ cells) resided mostly in L1 and deep layers and were less prominent in L2/3 (Supplementary Fig. 9). These results are in contrast to the pattern observed by Switch-ON injections (Fig. 7a, b) and by VGlut2 immunostaining (Supplementary Fig. 8), but consistent with the mutually exclusive distribution of VGlut1 and VGlut2 terminals in superficial and deep layers of the gRSC and the paucity of VGlut1-positive terminals in superficial layers[65]. To verify that these projections emerged from non-bursting cells, we recorded from subicular neurons in the presence of synaptic blockers to identify presynaptic neurons expressing ChR2-EYFP. Of the 23 subicular cells tested, 4/6 putative regular firing cells and 8/8 putative interneurons fired action potentials in response to light stimulation, but only 1/9 identified bursting cells responded to light (Supplementary Fig. 9). Overall, these experiments suggest a division of labor in the subiculum, whereby bursting VGlut2$^+$ pyramidal cells excite superficial gRSC and the VGlut2-negative cells drive predominantly inhibition in both superficial and deep gRSC cells. The source of the inhibitory drive is likely feedforward activation of inhibitory interneurons in gRSC, possibly complemented by direct long-range GABAergic projections[66–68], while the source of the residual excitation could stem from a sparse direct projection from CA1 pyramidal cells[69,70].

**Identification of subicular bursting cells in vivo**. Subicular bursting cells are strongly activated during SPW-R events[71]. The relation between burst propensity of neurons in vivo and SPW-R modulation was previously quantified in intracellular recordings based on the unambiguous bursting pattern following rheobase current injection[71]. However, this relation is less clear in extracellular recordings, where spike autocorrelograms (ACGs) are used to quantify the extent of complex spike bursts. Our aims here were twofold: first, to test whether the findings of Böhm et al.[71] using intracellular recordings in vivo and in vitro, can be extended to extracellular unit recordings and the emission of complex spikes, and second, to test the validity of the VGlut2-Cre

line as a reliable marker for bursting cells in vivo. We recorded from 196 subicular units in VGlut2-Cre mice ($n = 3$) injected with ChR2-EYFP (Fig. 8a) and used brief light pulses (10 ms) at random intervals throughout the recording session to identify VGlut2-positive neurons. Light stimulation increased unit activity in the subiculum, but not in CA1, confirming a specific localization of the construct to the subiculum (Supplementary Fig. 10a). Putative pyramidal cells were separated into light responsive and non-responsive groups (Fig. 8b top row and 8c). Only units that displayed a significant increase in firing rate within the 10-ms long light pulse were considered directly light activated. Inspection of individual and average ACGs of light responsive and non-responsive units suggested striking differences in burstiness (Fig. 8b bottom, 8d and Supplementary Fig. 10c). To quantify those differences, we applied a double exponential fit model to several key features of the ACG which are indicative for burst firing (Supplementary Fig. 10f). Fit parameters of responsive pyramids significantly differed from non-responsive cells in both rising slope, peak height and decay slope (Supplementary Fig. 10g), but not in baseline firing probability. Thus, light-responsive cells had a stronger tendency to be bursty, consistent with previous in vitro results showing that VGlut2 is a specific marker for bursting cells in the subiculum[59].

Next, we classified neurons according to their modulation by local SPW-Rs (Fig. 8b middle row and Fig. 8e). To avoid ambiguities, units that did not show significant up- or down-modulation of their firing rates by SPW-Rs were excluded from the analysis ($n = 43$). Ripple-excited units ($n = 84$) displayed more bursty autocorrelograms than ripple-suppressed units (Fig. 8b bottom row, 8 f and Supplementary Fig. 10d; $n = 26$). The majority (72%, 26/36) of light-responsive neurons belonged to the SPW-R—excited group (Supplementary Fig. 10e). It is noteworthy that these opto-tagged cells likely underestimate the VGlut2$^+$ population due to multiple factors such as transfection efficiency and reduced activation of cells on sites distal to the light source. In summary, these results validate the VGlut2 gene as a marker for bursting subicular cells and show that these neurons are excited by SPW-Rs.

**Subicular bursting neurons mediate SPW-R-related hippocampal output**. We next investigated whether the ripple coupling observed between the hippocampus and the gRSC can be mediated by VGlut2$^+$ subicular bursting cells. We simultaneously monitored dorsal subiculum and gRSC activity in VGlut2-Cre mice injected with ChR2 in dorsal subiculum, while optogenetically inducing high-frequency oscillations (100 ms sinusoidal or square pulses) in the subiculum ($n = 10$ sessions from four mice)[39]. Optogenetic stimulation of bursting subicular VGlut2$^+$ cells was sufficient to induce local ripple-like high-frequency oscillations (iHFO; Fig. 9a). Similar to the gRSC response to spontaneous SPW-Rs, iHFOs were associated with a negative wave and increased ripple band power in superficial, but not deep layers of the gRSC (Fig. 9b-d and Supplementary Fig. 11). The activity of light-responsive subicular pyramidal units identified using brief 5 ms light pulses was up-modulated around spontaneous gRSC ripple events and ramped prior to that of gRSC units (Supplementary Fig. 11). In many gRSC neurons, optical stimulation initially induced spike suppression, followed by rebound spiking approximately 100 ms after the stimulation onset (Supplementary Fig. 11). A similar temporal relationship was detected between subicular stimulation and cortical packets (Fig. 9e).

Finally, to directly test the contribution of VGlut2$^+$ subicular cells on gRSC activity, we simultaneously recorded from the CA1 and gRSC in VGlut2-Cre animals ($n = 5$) injected with an inhibitory optogenetic construct (AAV1-DIO-Arch-EYFP) into

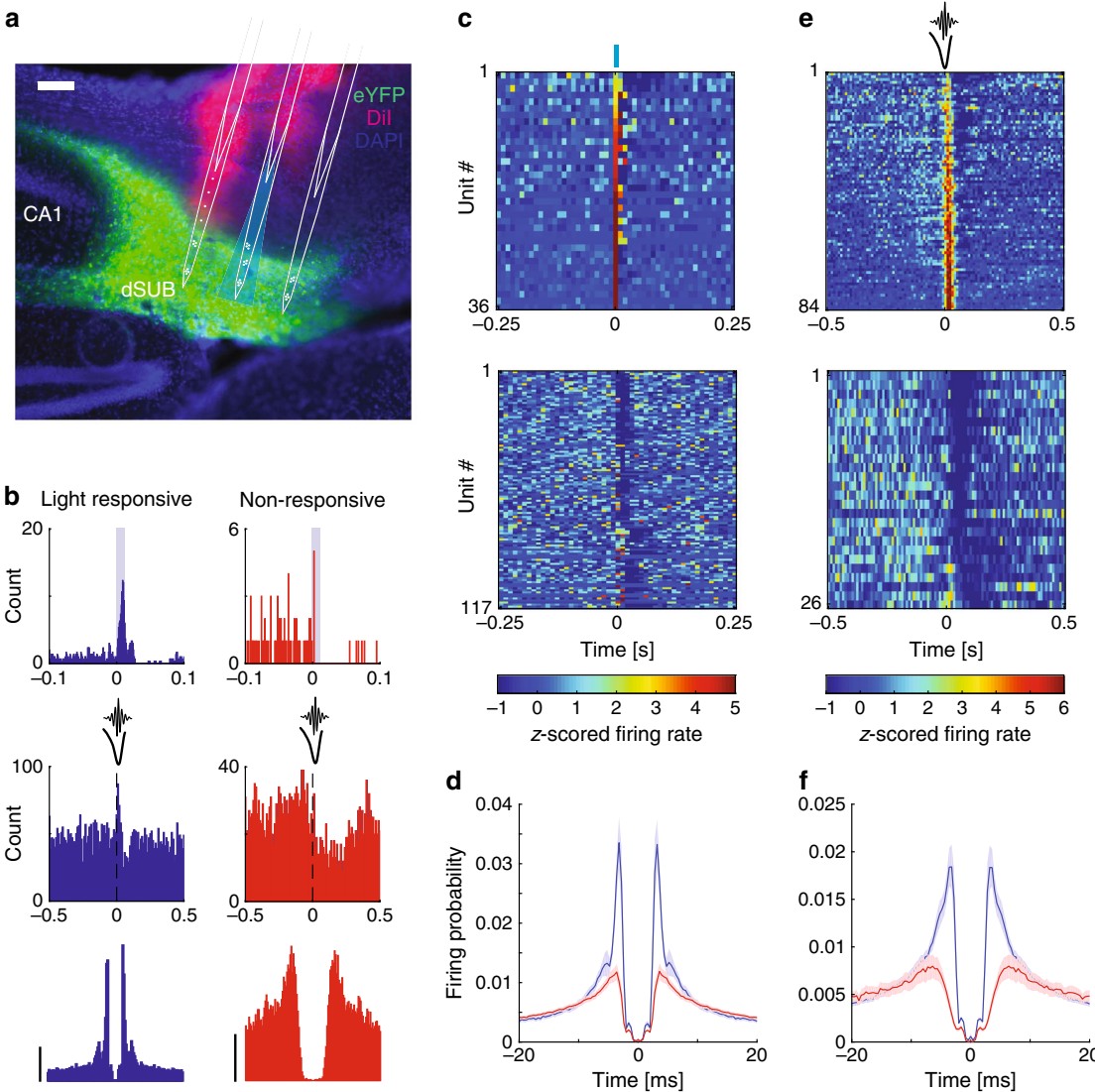

**Fig. 8 Identification of subicular bursting cells in vivo. a** Subicular bursting cells were labeled with ChR2-eYFP (green) using the VGlut2-Cre mouse line. Probes were coated with a dye (DiI, magenta) to confirm subicular targeting. Scale bar, 100 μm. **b** Examples of a light responsive (blue, top left) and a non-responsive (red, top right) subicular units, their responses to SPW-Rs (middle) and their ACGs (bottom). ACG range ± 25 ms; scale bar, 10 Hz. **c** Z-score normalized PETHs showing the responses of all units, which are positively (top) modulated or not positively modulated (bottom) within the 10 ms blue light stimulation (blue bar at 0 s, n = 6 sessions from three mice). **d** Average ACGs of light responsive (blue) and non-positively responsive (red) units. **e** Z-score normalized PETHs showing the responses of all the units, which are positively (top) or negatively modulated (bottom) by SPW-Rs (n = 6 sessions from three mice). **f** Average ACGs of positively (blue) and negatively (red) modulated units.

the dorsal subiculum. Green light stimulation (5 s pulses) of VGlut2+ terminals in the gRSC was contingent upon ripple detection in CA1, with a varying random delay to onset (mean delay 85.5 ± 0.8 ms). This protocol guaranteed that stimulation was delivered in brain states with high ripple occurrence rates (Fig. 9f). Inhibition of VGlut2+ terminals in gRSC significantly reduced the occurrence of gRSC ripples (Fig. 9g), despite a moderate increase in CA1 SPW-R incidence possibly caused by the bias introduced by the closed-loop protocol (Supplementary Fig. 11). Firing rates of individual gRSC units were also affected by inhibition of VGlut2+ terminals with the majority of modulated units (54/123, 44%) decreasing their activity and a small fraction of neurons positively modulated (12/123, 10%; possibly by disinhibition), suggesting complex microcircuit effects (Fig. 9h). To test whether the coupling of gRSC units to SPW-Rs was affected by the Arch-induced suppression, we computed the standardized cross-covariance between CA1 and gRSC unit pairs

(n = 391) obtained from a ±250 ms window around all SPW-R events occurring either within or outside of the stimulation. Average cross-covariance showed a significant reduction during stimulation, suggesting a decrease in excitatory drive (Fig. 9i). This finding was supported by a significantly decreased modulation index for ripples occurring during illuminated epochs (Supplementary Fig. 11).

## Discussion
We outline a functional pathway that supports the propagation of SPW-R activity from the dorsal CA1 to the gRSC via the subiculum. This pathway is routed through a specific subpopulation of bursty VGlut2-expressing cells, which project to superficial layers of the gRSC. Single neuronal assemblies in gRSC reliably distinguished between different constellations of hippocampal unit activity during SPW-Rs. Spontaneously occurring SPW-Rs,

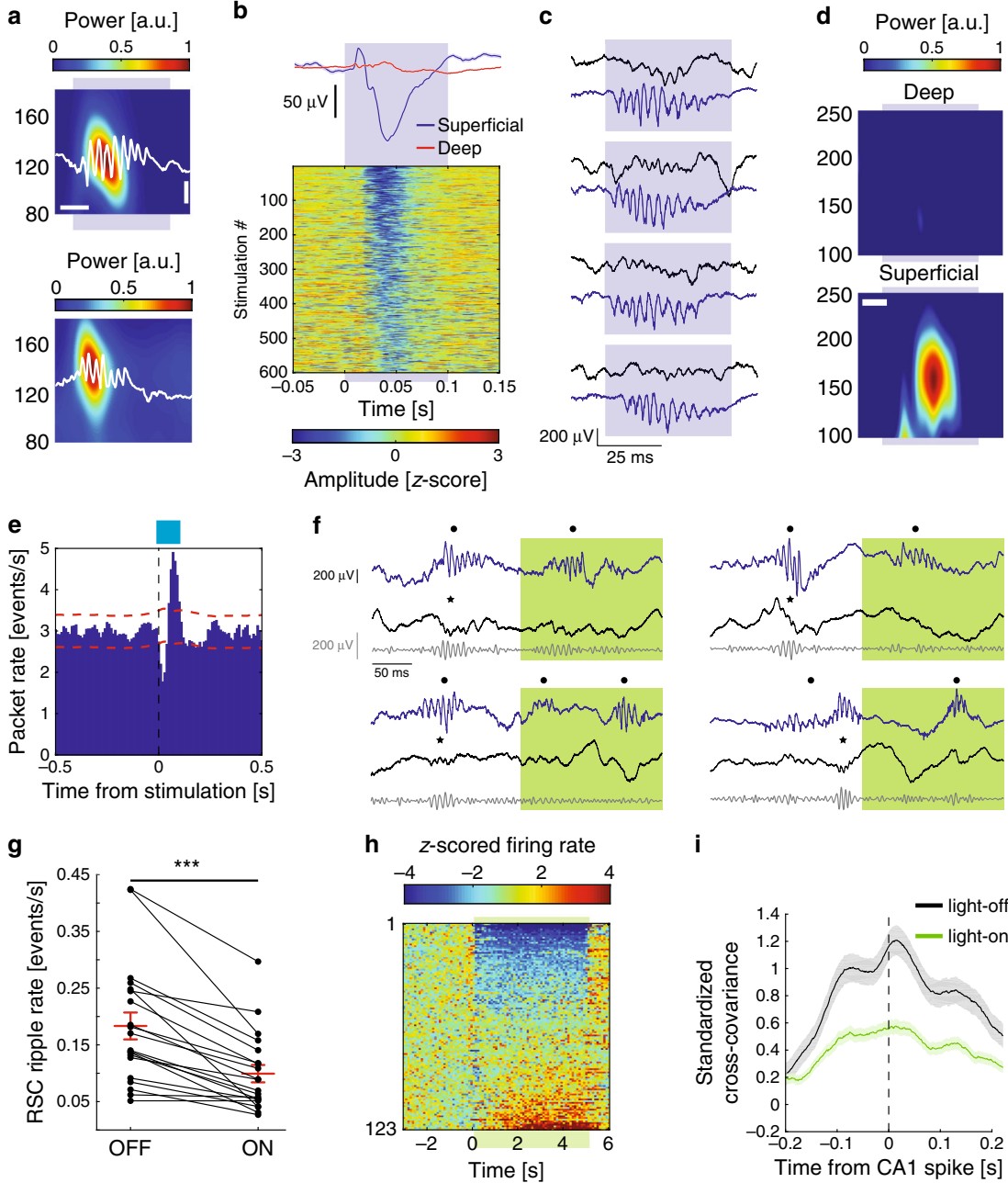

**Fig. 9 Subicular bursting neurons mediates SPW-R-related hippocampal output to the gRSC. a** Example normalized wavelet spectrograms of iHFOs (top, shaded blue area) and spontaneous ripples (averages of 155 iHFOs and 742 spontaneous ripples from one session). Example traces are overlaid in white. The *Y*-axis, frequency; Horizontal scale bar, 25 ms; Vertical scale bar, 200 μV. **b** Example z-scored voltage traces from a superficial gRSC channel in response to sine wave stimulation of dorsal subiculum (bottom) and the average response from a superficial (blue) and a deep (red) channel ($n =$ 2 sessions from one animal). **c** Example gRSC ripples (top, blue traces) induced by subicular stimulation (bottom, black traces). **d** Wavelet spectrograms from superficial (bottom) and deep (top) sites of gRSC centered around iHFOs induced in the subiculum ($n = 6$ sessions from four animals, 1,472 stimulations). Scale bar, 25 ms; same color bar; the *Y*-axis, frequency. **e** Cross-correlation of subicular VGlut2$^+$ opto-stimulation with gRSC packets ($n = 13$ sessions from four animals 5284 stimulations and 82,589 packets). **f** Example traces from a hippocampal channel (top, blue) and a superficial gRSC channel (middle, black), as well as the corresponding gRSC ripple-band filtered trace (bottom, gray) during and outside of green light stimulation (shaded area) of subicular VGlut2$^+$ terminals in gRSC. Black dots, SPW-Rs detected in CA1; asterisk, ripples detected in gRSC. **g** Optogenetic inhibition of subicular VGlut2$^+$ terminals decreased the rate of spontaneous gRSC ripples ($p = 0.0000885$, two-sided signed rank test; $n = 20$ sessions from four animals; 6487 ripples during light-off and 1068 ripples during light-on; mean ± s.e.m.). h) Z-score normalized PETH of all recorded gRSC units. Shaded green area denotes time of stimulation. **i** Standardized cross-covariance between CA1 and gRSC unit pairs ($n = 391$) from spikes occurring ± 250 ms around SPW-Rs either outside (black) or during (green) stimulation ($p = 0.0000002$, two-sided rank sum test; $n = 18$ sessions from five animals; see Supplementary Fig. 11).

optogenetic stimulation of pyramidal cells in the distal CA1 and subiculum, and optogenetic stimulation of VGlut2$^+$ neurons in the subiculum all induced population spiking and two forms of patterns in the superficial gRSC: a negative wave and

superimposed fast oscillation ('gRSC ripple'). In contrast, silencing VGlut2$^+$ neurons disrupted the spread of hippocampal SPW-R activity to gRSC. Hippocampal SPW-R to gRSC ripple coupling was state dependent and its occurrence was biased by

ongoing cortical population patterns, associated with power increase in the 3–10 Hz band.

Fast LFP oscillations typically arise in response to a strong excitatory drive in cortical networks[4]. Optogenetic excitation of cortical circuits gives rise to a fast oscillation (100-200 Hz), reminiscent of the physiological SPW-Rs in the hippocampal CA1 area[39]. In addition to the hippocampus, ripples have been described in the subicular complex, deep layers of the entorhinal cortex[16] and, more recently, in neocortical limbic/associational areas[17]. Here, we report LFP ripple activity in superficial layers of the gRSC. gRSC ripples reflected entrainment of superficial pyramidal neurons and putative interneurons. Thus, it is likely that population oscillations underlying LFP ripples are brought about by transiently increased drive of both excitatory cells and interneurons and their interactions in gRSC. This hypothesis is supported by the presence of a large-amplitude negative wave, shown as a sink by CSD analysis, in the superficial layers, upon which the ripple was superimposed, and the associated increased spiking of a large fraction of superficial gRSC neurons. This wave-ripple coupling is analogous to the SPW-induced depolarization of the apical dendrites of CA1 pyramidal cells and the coupled ripple in the cell body layer. Thus, the sharp wave and ripple reflect two distinct but often-coupled physiological events with homologous mechanisms in the hippocampus and gRSC. In contrast to superficial layers, ripple activity in deep layers of the gRSC was less prevalent, and the engagement of deep pyramidal units in ripple events was weaker. A number of factors could contribute to the propensity of ripple activity in superficial layers. First, as our in vivo and in vitro analyses suggest, hippocampal output, which acts as one potential drive of gRSC ripples, is most strongly concentrated at superficial gRSC layers. Second, superficial gRSC pyramids are more excitable than their deep peers, as implicated by their higher membrane resistance and low rheobase (Supplementary Table 1), and are therefore more likely to be recruited during ripple events. Lastly, as recent work suggests[72], superficial, but not deep layers of the gRSC are governed by strong feedforward inhibition, consistent with prevailing models of ripple emergence in the hippocampus[73].

A substantial portion of gRSC ripples co-occurred with hippocampal SPW-Rs (Fig. 2d–f and Supplementary Fig. 3), suggesting that an important source of excitation needed for gRSC ripples is the strong excitatory hippocampal outflow during SPW-R events. This unidirectional flow of activation is supported by multiple findings. First, gRSC ripples tended to follow CA1 SPW-R onset. Second, optogenetic induction of iHFO in the subiculum often induced gRSC ripples. Third, the pattern of activity during hippocampal SPW-Rs could be differentiated by gRSC neurons. Fourth, gRSC ripples and single units showed strong phase coherence with hippocampal SPW-Rs.

A fraction of gRSC ripples were not associated with a substantial increase in hippocampal ripple-band power (Fig. 2e). We cannot exclude that some of these events reflected localized undetected SPW-R output from sites more posterior to our recording sites in dorsal CA1 and subiculum[74]. Yet, the observation that gRSC ripples were often time-locked to 6–10 Hz oscillatory or intermittent events suggest that neocortical or, possibly, thalamic excitatory inputs may also contribute. We hypothesize that gRSC ripples emerge as long as superficial neurons experience sufficient level of excitatory drive, independent of the source of excitation. Correspondingly, the reduction in gRSC ripple rate observed during optogenetic inhibition may be caused by, in addition to the suppression of SPW-R-associated inputs, reduced tonic subicular input, which may lead to reduced excitability of superficial gRSC neurons and thereby a reduced likelihood of gRSC ripples, irrespective of the source of the afferent drive. We also acknowledge that pH and intracellular

$Ca^{2+}$ changes resulting from the proton pumping activity of Arch may have also contributed to changes in excitability[75,76]. Overall, our findings suggest that the gRSC is an important anatomical node for the dissemination of hippocampal messages to its wide neocortical partners during SPW-Rs.

We present evidence that stimulation of VGlut2$^+$ subicular bursting cells is sufficient to phase-reset activity packets in the gRSC (Fig. 9e), thereby influencing awake cortical population activity. These data suggest that, similar to NREM UP and DOWN states (or perhaps as an awake variant thereof), activity packets represent a metastable regime, and that any perturbations to it will result in a state transition as long as they are sufficiently strong[77–79]. Consequently, differences in cortical responses to spontaneous SPW-R and subicular stimulation may reflect the fact that stimulation did not always occur at times permissive for optimal cortical responsiveness and efficient hippocampal-cortical cooperativity. Altogether, these data implicate the subicular VGlut2$^+$ bursting cells as a boosting mechanism in hippocampal – cortical communication by means of maintaining cross-structural synchrony.

Responses of gRSC neurons to SPW-Rs were multiphasic and exhibited a short latency peak surrounded by a pre-SPW-R and a delayed peak ~150 ms before and after SPW-R onset, respectively, suggesting that SPW-Rs are embedded within more complex cortical dynamics oscillating at around 6 Hz. This cortical rhythm was associated with both increased gamma and multi-unit activity, primarily in deep layers (Fig. 6)[80]. While, numerous studies have reported the coupling of hippocampal SPW-Rs to cortical rhythms such as slow-waves or spindles[15,21,36,46,81,82], the precise contribution of hippocampal output has been difficult to disentangle in the absence of known monosynaptic connections in anatomically and molecularly defined cortical subpopulations. As a consequence, cortical activity that is temporally correlated with hippocampal SPW-Rs is often interpreted as being directly driven by the hippocampus, thereby neglecting the contribution of locally generated cortical currents. For instance, cortical cells are differently modulated by SPW-Rs, depending on whether SPW-Rs coincide with UP- > DOWN[8,70,77] or DOWN- > UP[17,36] transitions. These fine timescale dynamics are likely to be area and brain state dependent[79]. Here, we report that deep gRSC pyramids show weak phase locking to hippocampal ripples and a lack of prominent short latency excitatory responses to subicular drive both in vitro (Fig. 7h) and subicular or CA1 drive in vivo (Figs. 3a, d and 9), suggesting that these cells are more likely recruited and modulated by cortical packets rather than by the SPW-R themselves. In contrast, the additional SPW-R-locked activity seen mainly in superficial layers and interneurons is suggestive of direct ripple-associated currents. Consistent with this view, superficial gRSC pyramidal cells receive monosynaptic inputs from the subiculum (Fig. 7) and show rate and phase locking to CA1 SPW-Rs (Fig. 3a, d). These findings suggest that while the hippocampal and cortical networks are globally coupled, only certain populations of cortical neurons are tuned to hippocampal inputs, while others are engaged in local cortical dynamics.

In conclusion, our findings demonstrate that, in addition to the hippocampal-entorhinal and hippocampal-prefrontal routes, the hippocampal-subicular-gRSC path is an alternative channel to broadcast hippocampal content. In addition, our findings suggest that the mechanisms of SPW-Rs in the hippocampus apply to the retrosplenial cortex as well: a synchronous, anatomically concentrated excitatory drive (reflected in a sharp wave) enables a state change and triggers high-frequency oscillations, supported by fast spiking GABAergic interneurons. It remains to be clarified how ripple-associated information routed to the gRSC differs in its content from spike information sent to other areas, and how

the transformed spike pattern at each synapse contributes to memory. By elucidating a specific anatomical pathway linking hippocampal SPW-Rs to gRSC activity, involving a genetically-defined subpopulation of bursting cells in the subiculum, we provide concrete targets for future studies addressing these questions.

## Methods

**Animal research.** Animals were colony housed in a 12-hour light/dark cycle with free access to food and water (ambient temperature: 21°C; humidity: 54%). All experiments were conducted in accordance with European guidelines and with permission from local regulatory authorities (Berlin Landesamt für Gesundheit und Soziales, permits G0092/15 and G0150/17 and the Institutional Animal Care and Use Committee of New York University Medical Center). Total numbers of animals used for this study were as follows: eight wild type C57BL/6 mice (JAX: 005304), 24 transgenic VGlut2-Cre mice (JAX: 016963) and three transgenic CaMKII-Cre::Ai32 mice (JAX 005359 with JAX 012569).

**Stereotaxic surgery and viral injections.** Mice were anesthetized under isoflurane (1.5% vol/vol in oxygen) and body temperature was maintained at 37°C. The coordinates for targeting the subiculum and the gRSC were as follows (from Bregma, in mm): −3.0 AP, ±1.6 ML, −1.6 DV and −2.7 AP, ±0.75 ML, −1.0 DV, respectively. A NanoFil syringe with a 34-gauge needle (with UMP3 microinjection system and Micro4 controller; all from WPI Inc. Sarasota, USA) was used to inject 100 nL of AAV virus with various constructs (Charité viral core facility; titer $1.3\times10^{11}$–$1.6\times10^{12}$ VG/mL) at a rate of 20 nL/min, waiting 5 min after each injection before slowly retracting the needle. Animals were allowed to recover for at least 4 weeks before the experiment to allow for a proper expression of the construct at the axon terminals. For in vivo head plate implantation, the craniotomy site (AP: −2.5–3.0, ML: 0.5–2.5) was marked, and a stainless steel ground screw was placed in the contralateral frontal bone, or, in the transgenic CaMKII-Cre:: Ai32 mice, a stainless steel wire was implanted intra-cranially above the cerebellum. A custom-made metal lightweight head holder was attached to the skull using adhesive cement (Paladur; Heraues, Germany). After surgery, mice were returned to their home cage and allowed to recover for a minimum of 2 days before habituation started. Habituation in the recording setup was repeated for up to one week in increasing increments of fixation time until the animal sat quietly for at least 1 h. On the first day of an experiment, mice were anesthetized and a craniotomy was made above the left gRSC and dorsal CA1 or subiculum and covered with silicon sealant (Kwik Cast, MicroProbes). Mice were allowed to recover for at least 3 h before the experiment. Before the recording started, Kwik-Cast sealant was removed and the brain was washed with ACSF. An acute silicon probe (A4x8-5mm-100-400-703-A32; Isomura32; ISO-3×-tet-lin; A8x32-Poly2-5mm-20s-lin-160; Neuronexus, Ann-Arbor, MI, USA), in many cases covered with a fluorescent dye to enable post-hoc track identification (DiI, DiR or DiO, Thermo Fisher Scientific, MI, USA), was lowered slowly into the brain at a 30° angle (coronal orientation) until ripples were clearly detected on the hippocampal shanks (~1.5–1.7 mm DV). Recordings started 15-20 min after the probe reached final depth and lasted approximately 60 min. After the end of the recording, the probe was slowly retracted, the brain washed with ACSF and covered again with Kwik-Cast sealant.

**Acute slice electrophysiology.** Mice ($n = 12$ VGlut2-Cre mice; 11 males, 1 female; age: 2– months) were deeply anesthetized with isoflurane, decapitated and the brains removed. Tissue blocks were then mounted on a vibratome (VT1200S, Leica Biosystems, Wetzlar, Germany), and slices containing either the subiculum or gRSC were cut at 300 μm nominal thickness. The slices were stored in an interface chamber where they were stored for 1–5 h before being transferred to the recording chamber where they were perfused at a rate of 3-4 ml/minute. The recording chamber was mounted on an upright microscope equipped for IR-DIC microscopy. Whole-cell recordings were performed using a Multiclamp 700 A (Axon Instruments, CA, USA) using glass microelectrodes filled with 120 mM K-gluconate, 10 mM Hepes, 3 mM Mg-ATP, 10 mM KCl, 5 mM EGTA, 2 mM MgSO₄, 0.3 mM Na-GTP, 14 mM phosphocreatine and 2 mg/mL biocytin. The resistances of the electrodes ranged between 3 and 5 MΩ. Access resistance (< 20 MΩ) was continuously monitored during the recording and was not allowed to fluctuate by more than 20%. Data were analyzed online using Igor Pro and offline using Matlab (Matworks) and were not corrected for liquid junction potential. Light-evoked responses were detected if the peak of the postsynaptic potential averaged over >10 repetitions crossed ± 0.4 mV.

**Histological processing.** At the end of the in vivo experiments, animals were deeply anesthetized using urethane (2.5 g kg⁻¹ body weight) and transcardially perfused with phosphate-buffered saline (PBS), followed by 4% paraformaldehyde (PFA) in PBS. Following an overnight fixation in PFA, brains were carefully washed in PBS before they were mounted on a vibratome (Leica VT1000S, Leice Biosystems, Wetzlar, Germany) and cut into 100 μm slices. Sections were mounted in either Vectashield (Vector laboratories) or DAPI-containing Fluoroshield

(Sigma-Aldrich). For in vitro experiments, slices were transferred to a PFA solution and fixated overnight. Immunoreactions for GAD-67 were carried out using primary mouse antibody (diluted 1:500, MAB5406, Millipore) and secondary Alexa 555 (1:500, A-21424, Invitrogen). Streptavidin was conjugated to Alexa 647 (1:500, S32357, Invitrogen) for visualizing the biocytin. Sections were then mounted in Fluoroshield and imaged using a Leica TCS SP5 confocal microscope (Leica Biosystems, Wetzlar, Germany). Images were analyzed using the free software ImageJ. The location of individual cortical recording sites was assigned to either deep or superficial layers based probe tracks and recording depth

**Extracellular data preprocessing.** Signals were acquired using an RHD2000 system (Intan Technologies, LA, USA) at 20 kHz and resampled at 1.0 or 1.25 kHz using a low-pass sinc filter with a 450 Hz cut-off band to extract the LFP data. Spike clusters were extracted from the high-passed filtered and channel whitened signal using Kilosort[83]; a manual curating step, where units were merged based on common refractoriness and waveform similarity was performed using Klusters (http://neurosuite.sourceforge.net/)[84]. Units with firing rate < 0.5 Hz or ISI violation > 0.01 were discarded. Units were separated into putative pyramids and interneurons based on the spike width, waveform asymmetry and firing-rate using k-means[34].

**In vivo data analysis.** Statistics: Data were analyzed using Matlab (Statistical Toolbox; Buzcode Toolbox). No statistical methods were used to pre-determine sample sizes, but sample sizes chosen are similar to those from other publications in the field. Unless otherwise indicated, data are displayed as mean ± s.e.m and statistical significance is deemed using two-sided Wilcoxon Rank-Sum test.

Ripple detection: For offline ripple detection, the channel with the largest ripple amplitude was selected by visual inspection and the LFP signal was filtered at 100–280 Hz in forward and reverse direction using a 2nd order Butterworth filter. The signal was then rectified, and smoothed using a Savitzky-Golay filter. Ripple events were defined as events where the smoothed rectified signal was more than 2 SDs and the filtered signal more than 4 SDs above the respective mean for at least 20 ms. In the case of two events separated by less than 40 ms (peak to peak), the event with the smaller amplitude was discarded. Detections were visually inspected using Neuroscope (http://neurosuite.sourceforge.net/).

SPW-R and stimulation modulation: Only sessions with > 100 ripple events selected for analysis. Peri-event time histograms (PETHs) and cross-correlograms were constructed using SPW-Rs separated from each other by at least 500 ms to avoid contribution from ripple bursts[85]. Significance of SPW-R and stimulation PETHs was calculated using an adapted convolution method[86].

Spectral analysis: Spectrograms and phase coherograms (Fig. 2g) were constructed using a complex Morlet wavelet convolution with logarithmically spaced number of cycles[87] and are either plotted on a logarithmic scale or z-score normalized relative to a baseline period at least 100 ms prior to SPW-R/ stimulation onset. Power coherograms (Supplementary Fig. 3) were constructed using the Matlab function 'wcoherence'. Spike-LFP coherence was computed using a multi-taper based analysis (Chronux)[88] with five tapers and a time-bandwidth product of 3. Phase was extracted using the filter Hilbert method. Only units significantly modulated by SPW-Rs (with a significant phase preference, defined as $p < 0.05$ and ITPC > 0.1) on Rayleigh's test of circular non-uniformity were selected for analysis. Inter-trial phase clustering (ITPC) was computed as $\text{ITPC}_{tf} = |n^{-1}\sum_{r=1}^{n}e^{ik_{tf}}|$, where $n$ is the overall number of ripples and k is phase angle at ripple number $r$ and time frequency point tf, which was set to be the trough of the ripple[87].

Identification of negative waves: Negative waves were identified in a superficial channel chosen by visual inspection using a template matching algorithm defined based on visual selection of identified events in Clamp fit 10.4 (Molecular Devices, CA, USA) and thresholded at 1 RMS.

CSD analysis: CSD was estimated using the inverse method[89]. Analysis was performed on an average of > 150 evoked potentials from one shank in 8 or 15 cortical depths after subtraction of the shank average.

Phase-amplitude coupling: The coupling between high-frequency amplitude and lower frequency phase was estimated using a modulation index[90]. Phase was calculated at 39 frequencies from 2–40 Hz and power was calculated at 91 frequencies from 50 to 500 Hz with wavelet convolution. Phase time series were binned into 50 intervals and the mean wavelet amplitude at each bin was computed. The modulation index was extracted by measuring the divergence of the observed power distribution from the uniform distribution. Significance of MI values was deemed by creating 200 randomly shifted surrogates for each power frequency and computing a $P_z$ value. Non-significant values ($p < 0.05$) were set to zero.

Independent component analysis (ICA): ICA-based separation of voltage signals was performed using the function 'jader' from the EEGLAB toolbox (https://sccn.ucsd.edu/eeglab/index.php).

Cross-covariance estimation: The cross-covariance was calculated as previously described[38]. In short, the cross-covariance estimate was first computed as CCE = $\frac{CCH}{b*T}$ − $R1*R2$ with CCH as the cross-correlation histogram $b$ the bin size, $T$ the period of observation and $R1$ and $R2$ as the firing rate of a hippocampal and a retrosplenial unit, respectively. The standardized cross-covariance was then computed as CCE* $\sqrt{\frac{b*T}{R1*R2}}$.

Packet detection and state-index definition: Activity packets were defined as local maxima in the smoothed MUA population vector (Gaussian SD: 30 ms) that exceeded the local average (in a 10 s window) for at least 25 ms by at least 60% and were surrounded by local minima (<20% of the local average) for at least 25 ms. Events with inter-packet interval below 15 ms were merged together. Packets were only identified in sessions with >15 units. Accurate detection was visually inspected using Neuroscope. The state-index (SI) was computed by shifting a moving window (0.5 s) along the smoothed MUA firing rate (step size: 1 ms) and counting the proportion of nonzero time bins[51].

Electromyogram (EMG) signal was extracted as previously described[55,91]. In short, the LFP from multiple recording sites was filtered between 300-600 Hz and the zero time lag correlation coefficient between them was computed over a sliding window of 0.5 s.

Clustering of hippocampal SPW-Rs: SPW-Rs were detected as described above and the normalized firing rate of dorsal hippocampal neurons was calculated for a single 100 ms time window after ripple onset. Fast-spiking interneurons were excluded from these firing rate vectors. K-means clustering (k = 10, centroid initialization, and squared Euclidean distance) was then used to categorize CA1 ripples according to each 10-dimensional vector representation.

RSC discrimination of hippocampal SPW-R type: Peri-event time histograms (10 ms bins) of RSC activity was constructed around CA1 ripples. Ripple type discrimination was assessed using an ANCOVA, where the groups were CA1 ripple cluster ID, and the continuous covariate was the mean population firing rate in CA1, used to control for the fact that more RSC neurons fire after larger CA1 population bursts. A group main effect of p < 0.01 was used as threshold for ripple type discrimination. In addition, the labels of the ripple clusters were shuffled and the observed F-ratio from the ANCOVA test was compared to the one derived from the shuffle distribution.

RSC LFP discrimination of hippocampal SPW-R type: Spectrograms were computed using wavelet convolution of the down-sampled LFP using a modified Morlet kernel with 80 logarithmically spaced frequency bands from 5 to 300 Hz. Frequency amplitudes were z-score normalized relative to the entire ±500 ms observation window. Using a similar model as that used for RSC units, an ANCOVA was used to determine whether energy at a specific frequency band at a specific time around ripple onset distinguished ripple type, while regressing out the effect of CA1 population firing rate. The p-values describing the main effect of ripple cluster ID were combined across subjects ($n = 3$ mice) using Fisher's method $-2^{*}\sum_{i=1}^{n}\log p_i \approx \chi^2(2^{*}n)$, where the global p-value was taken from the $\chi^2$ distribution with 2*n d.f.

t-SNE analysis of hippocampal ripple clusters: For each session, each ripple was defined by two population firing rate vectors: one for the hippocampus and one for gRSC. The hippocampal population vector was defined as above, while the firing rates for the gRSC were taken over the 200 ms after hippocampal ripple onset. T-SNE[92] was calculated separately for each region using perplexity = 30 and initialized according to[92] (i.e. by sampling map points randomly from an isotropic Gaussian around the origin).To quantify the clustering of gRSC ripple activity in this lower dimensional space, the T-SNE coordinates were binned (bin size = 1 arbitrary unit) and smoothed (2D Gaussian, σ = 4 AU) and normalized by the number of ripples within each cluster. The observed density was compared against a shuffle distribution ($n = 1000$) in which the labels were randomized.

Optogenetic manipulations in vivo: For optogenetic manipulations in CamKII-Cre::Ai32 animals, (F1 generation of breeding JAX # 005359 with JAX # 012569; three adult females), we used either a 256 channel A8x32-Poly2-5mm-20s-lin-160 Neuronexus probe with optic fibers (50 μm core diameter) mounted on each shank and coupled to a laser diode (Thor Labs, NJ, USA) or a 32-site silicon probe with integrated μLEDs (Neurolight, MI, USA). For optogenetic manipulations in VGlut2-Cre animals (JAX: 016963, 2–3 month old, 8 males and 3 females), we used 32 channel ISO-3×-tet-lin probe (Neuronexus, MI, USA) mounted with an optic fiber (50 μm core diameter) coupled to a LED light delivery system (Plexon Inc., TX, USA). Photostimulation consisted of 5–10 ms (for tagging experiments) or 100 ms (for iHFOs) 473 nm and 5 s 555 nm light pulses. Measured light power at the probe tip ranged between 350 and 750 μW for 473 nm and 150 μW for 532 nm. Closed-loop stimulation was accomplished by filtering a manually selected hippocampal channel in the ripple band, on which light stimulation was triggered by crossing of a manually set threshold (Power 1401, Cambridge Electronic Design Limited, UK).

**Reporting summary**. Further information on research design is available in the Nature Research Reporting Summary linked to this article.

## Data availability
Data can be made available upon request.

## Code availability
The majority of the code used for this study was adapted from the buzcode repository (https://github.com/buzsakilab/buzcode) and the FMAT toolbox and is available under http://fmatoolbox.sourceforge.net/.

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

## Acknowledgements

The authors would like to thank Dr. Ofer Yizhar for providing the Cre-Off virus and Zachery Saccomano for providing software used in this study. We thank Dr. Nikolaus Maier and Constance Holman for insightful comments on earlier versions of the manuscript. We also thank Susanne Rieckmann and Anke Schönherr for excellent technical assistance. We thank SciDraw (Jason Keller) for the mouse brain schematic. This study was supported by grants from the Deutsche Forschungsgemeinschaft (DFG, German Research Foundation): Exc 2049, SFB 1315 (Project-ID 327654276), SPP 1665; the BMBF: 01GQ1420B, the National Institute of Health (U19NS104590), and a BrainPlay-ERC-Synergy grant. S.M. was funded by NIMH, K99 MH118423. P.B. was supported by a BIH Delbrück Fellowship (Stiftung Charité).

## Author contributions

N.N., S.M., and D.S. designed experiments; N.N., S.M., P.B., S.O., and D.F.N. performed the experiments; N.N. and S.M. analyzed the data; N.N., S.M., and G.B. wrote the paper with inputs from all authors; J.J.T. supervised the initial in vivo experiments; D.S. supervised the project.

## Competing interests

The authors declare no competing interests.
