## [Peer Review File · Nature Communications]

Reviewers' Comments:

Reviewer #1:

Remarks to the Author:

Nitzan et al. study the pathway underlying the propagation of ripples from CA1 to cortex. They confirm that the anatomy and connectivity previously described from the CA1 region of the hippocampus to the retrosplenial cortex (via the subiculum) is capable of relaying CA1 ripples to the cortex. Ripples are most likely to be seen in the superficial retrosplenial cortex, but are weaker in the deeper layers. They show that optogenetic activation of CA1 and subiculum can lead to ripples in superficial RSC. Importantly, they show that bursty VGLut2+ neurons in the subiculum are the ones most likely to relay this information. The manuscript represents a substantial amount of solid work and is not far from being ready for publication. It is also impactful and timely, as both the subiculum and retrosplenial cortex are crucial pathways in the flow of information from the hippocampus to the rest of cortex and are rightly receiving a lot of attention right now in the learning and memory field. The manuscript is appropriate for the journal and will be of interest to the field. However, there are some moderate changes that will help to clarify and improve the manuscript:

1. The authors show that a large proportion of retrosplenial ripples cortex are independent of CA1 ripples ("71% of gRSC ripples did not occur within ± 25 ms around dorsal hippocampus ripples"). Therefore in Figure 2b, it would be important to include a clear, simple bar graph showing what percentage of RSG-ripples were coincident with CA1 ripples and what proportion were not, in each animal recorded, as well as the average across animals. This is an important point that is made in the text and should be relayed in this figure as well.
2. In all cases, it appears that "deeper" layers of retrosplenial cortex are also more dorsal because of the steep angles of the probes. This is unavoidable with such medial recordings, but the authors should point out that the "superficial" arrows in most figures also represent "more ventral" locations and discuss in the text how dorsal the "deeper" layer contacts are compared to the "superficial" contacts in each animal and how this could impact interpretation of the data.
3. The CSD plot in Figure 1D is an important example. In the supplements, the authors should show this for each of the animals.
4. Figure 3 and Supplementary Figure 3 use different notations for the intermediate part of the proximodistal CA1 axis. "Intermediate" should be used throughout instead of the more confusing "medial" used in Figure 3.
5. On the same topic, the direct (local) response to hippocampal opto stimulation CamKII mice was largest in the proximal area. Given the differences in local response magnitude, the data in Figure 3c regarding RSC responses should be shown both before and after normalization by the local response magnitudes.
6. Very few units in RSC increase their firing rates in response to DH stimulation (Fig. 3E). This should be reflected in the Results describing this data and by a simple bar graph clarifying the proportion of neurons excited or inhibited in both superficial and deep layers. The group averages and even the color-coded rasters can make this information hard to discern.
7. The manuscript focuses on a pathway from CA1 to subiculum to retrosplenial cortex. For this reason using DH (Dorsal Hippocampus) as an umbrella term for CA1 and subiculum is not appropriate and can lead to confusion. The authors need to avoid using DH altogether in this manuscript, and instead use CA1 or subiculum to describe the precise area or sharp-wave ripple they are describing. This is

important for the clear interpretation of the results by all readers.

8. Page 2: "The spiking of putative interneurons lagged behind the trough-locked pyramidal neurons by 1 to 2 ms" – in terms of circular variables the "lagged behind" statement is not necessarily true, as there could be a 300 degree phase delay of excitatory neuronal firing during the hyperpolarization induced by the inhibition. This should be clarified.

9. Additional discussion on the role of VGluT1+ positive cells in the subiculum regarding information encoding and propagation would be valuable given the differences reported here. This should be added to the "Bursting pyramidal neurons..." section in the discussion.

Minor change:

1. On page 6: "Likewise, light stimulation of fast spiking interneurons in superficial layers" This makes it sound like the authors are stimulating directly expressing fast spiking neurons. Instead they are stimulating afferent fibers that express opsins while recording from fast spiking neurons in this case. This wording should be revised to make the recording and stimulation configuration clearer.

Reviewer #2:

Remarks to the Author:

This interesting paper reports ripple-like oscillations in RSC layer 2/3 coupled to hippocampal sharp-wave ripples. While the rate of RSC ripples is lower and some are uncoupled to hippocampal events, they correlate with cortical activity packets and negative waves, suggesting embedding dynamics during synchronous 6Hz cortical states. The paper describes the contribution of generic cell-types (PC and interneurons) and look to establish more direct links by manipulating hippocampal and subicular cells with optogenetic tools. Authors suggest that a subset of SUB cells (vglut2+) may play roles in relaying SPW-associated dorsal hippocampal firing to RSC .

The ms is well written, data are overwhelming and results are thought-provoking. I have however a couple of comments regarding specificity that require major clarification.

Major comments:

- Given SPW-r are relatively coherent along the entire hippocampus, it is difficult to establish the specific propagation route. Similarly, given the effect of global brain dynamics and state transitions, it is not surprising to find functional correlation between structures and regions during synchronous brain states. The challenge is to show that some RSC responses associated to dorsal CA1 SPW-ripples are running through or depend on vglut2+ SUB cells. While authors tried to establish the link, the ms reads more like two separate pieces; Fig.1-5 address functional correlations between DH and RSC, while Fig.6-8 look at the effect of manipulating subsets of SUB cells in RSC activity. Data in Fig.7 show activity of optogenetically tagged SUB cells during the local ripple, but not their specificity during RSC ripples coupled/uncoupled to CA1 events. Data in Supp.Fig.8e show that from the total of SPW positively modulated SUB cells (26+58=84) only few were vglut+ as judged by optotagging (26/84= 30%). The ms will strongly benefit from reinforcing specificity of vglut2+ SUB cells in coupling hippocampal activity to RSC during some SPW ripples. I guess authors can exploit their data to stress this point further.

- Similarly, specificity of optogenetic evidence should be more carefully addressed. First, there is mismatch between the RSC LFP response after CA1 stimulation (Fig.3b,d) versus vglut2+ SUB stimulation (Fig. 8c). While ripple-like oscillations are evoked all along the 100 ms light pulse in CA1,

only an early wave is evoked in response to SUB stimulation. It is unclear whether the high frequency spectrum shown in Fig.8d reflects more spiky wave components than a real LFP ripple oscillation. Second, RSC unit responses differ in CA1 (Fig.3e) versus vglut SUB stimulation (supp.fig.9a) in terms of rate, tonic firing dynamics and rebounds. Finally, the dynamics of SUB induced packet rates (Fig.8e) does not match with that of spontaneous ripples (Fig.5d) suggesting there is an additional delay when SUB cells are directly activated. It is therefore unclear whether optogenetic induced activity at CA1 and SUB is targeting RSC through different pathways.

Other comments:

- In their analysis, data from CA1 pyramidal layers and SUB are pooled collectively into DH (unclear contribution), but these regions are different in terms of PC identity and connectivity with RSC (thus part of heterogeneity may be region-specific). Authors may reinforce direct links between the dorsal CA1 and RSC "via" SUB by identifying CA1, SUB and RSC responses separately. This will help addressing my first major point.
- Authors use a burstiness index to evaluate the firing autocorrelogram in vivo and to presumably establish links with in vitro data. SUB bursting cells are defined by their response to current injection in vitro. Evaluation of complex spike behavior by the firing autocorrelogram is not directly related with the intrinsic bursting phenotype. Complex spikes are dendritically generated and obey to different mechanisms. For instance, CA1 pyramidal cells fire regularly when tested in vitro but exhibit complex spikes in vivo. This should be clarified to avoid confusion.
- Difference in burstiness of RSC-ACC cells and their potential association with hippocampal ripples was previously reported by Wang and Ikemoto (ref 17).
- Hippocampal iHFO looks more like pathological population spikes than physiological ripples (the amplitude, firing rate, spectral leakage of iHFO events in Supp.Fig 3b) suggest that high intensity optogenetic stimulation of CA1 in CaMKII animals is eliciting hypersync responses. Can more physiological events induced with lower light intensity? Please, clarify this point.
- While authors acknowledge the potential confounding effect of direct GABAergic CA1 projections to RSC (ref 67) their real impact in spontaneous and induced RSC oscillations is not addressed (e.g. by some pharmacological experiment). Similarly, optogenetic stimulation has direct and indirect microcircuit effects which can explain part of heterogeneity and unspecific responses. Any additional experiment or analysis addressing these issues will make the paper stronger.
- Regarding optogenetic tagging of vglut2+ : units inhibited by light stimulation reflect microcircuit effects; it is not necessarily indicative of a vglut2- phenotype.
- Arch experiments: why was a 50 ms delay chosen? While there is a significant reduction of RSC ripple reported in Fig.8g, raw data in Fig.8f raise doubts on how was light stimulation triggered and evaluated. First, the hippocampal ripple preceding light occurred more than 50 ms before, so it is unclear how is delay defined. Second, the ripple power is very low as compared with those following so it is unclear how was closed-loop defined. Third, while the power of the RSC ripple-like events is reduced by green light, multi-unit activity survives and some units are actually activated (possibly by disinhibition) as shown in Fig.8h. All this suggest complex microcircuit effects underlying hippo-RSC communication.
- Remondes and Wilson reported local RSC gamma oscillations coupled to hippocampal SPW ripples (though at more rostral levels). This is somehow suggested by spectrograms in Fig.5 and low

frequency components in Fig.2g. It is also suggested by the modulation of PETHs shown in Fig.2h. Please, address and clarify.

- page 7, section Identification of SUB "... striking differences in burstiness (Fig.5b bottom, 5d...)" should be Fig.7b,d. Thus inter regional coordination may be running through gamma and not only during ripples. Please, specify panel numbers when referring to Supp.Fig.8 in this section.

- terminology is not homogenous along the ms, which complicates reading. For instance, in sup.fig8e positively or negatively modulated refer to SPW and responsive/not-responsive refers to light, while in supp.fig.9b modulation refers to light.

- page 2, CSD map (Fig.1d) not e. Also, the probe shank is not parallel to the somatodendritic axis which may complicate interpretation of CSD. This is alleviated by using ICA, but worth noting.

- Sup.Fig.2 is called before Sup.Fig.1

- Methods should be more detailed. For instance, event detection (SPW, ripples, negative waves, packets) is dependent on a threshold which should be more carefully addressed. Event clustering is implemented by k-means with k=10 without justification.

- Clustering SPW analysis is very interesting (Fig.4). However, statistical significance of cluster specificity should be challenged with some additional methods and incorporated into the figure. Some rasters appear only marginally significant.

- Interneurons contribute heterogeneously to ripples both in hippocampal and cortical regions (Klausberger & Somogyi; Averkin & Tamas). While criteria for unit separation do not allow to disambiguate (see heterogenous contribution in Fig. 1g, 2i,j) it may be important to make a note on this point.

- Some terms and analyses are not standard and the general reader may have difficulties. The narrative will improve by including short descriptions to facilitate interpretation. For instance, cross-covariance.

- Fig.3: Distal optogenetic stimulation of CA1 may work better than intermediate or proximal stimulation due to off-target direct illumination of RSC terminals, especially because strong intensity was used. Can authors discard such an effect?

- check spelling and some typos (e.g. electromyographic..)

- Fig.7b, please add labels to facilitate reading. Also, caption identifies blue cells as bursting and red as regular-spiking cells without a clear criterion.

Reviewer #3:

Remarks to the Author:

This manuscript addresses the propagation of sharp wave-ripple (SWR) complexes to the retrosplenial cortex. A combination of In vivo (including optogenetics manipulations) and slice experiments are used to characterize the functional pathway from the hippocampus to the retrosplenial cortex during SWR in mice.

SWR are considered instrumental for the two stages model of memory trace formation as retrieval, which both require the broadcast of these highly synchronous events to cortex (e.g. to induce long-term potentiation in cortico-cortical connections). Evidence for the relationship of SWR to cortical activity can be found in the literature, and it has been noticed in particular that hippocampal SWR coincide with cortical SWR in several associative areas, including the retrosplenial cortex (Khodagholy et al. 2017). Besides this study, SWR in the retrosplenial cortex have not been investigated, as far as I know, although the contribution of this region to memory functions has been supported by multiple studies.

Content of the study

The experimental result demonstrate a very good command of electrophysiology and optogenetic techniques leading to high quality data. The data analysis is well executed. The study reproduces many aspects of the previous literature, witnessing in particular the occurrence of cortical SWRs with clear electrophysiological properties, both at the level of the LFP and spiking activity (Fig. 1). As also previously reported, a fraction of these SWRs cooccur with hippocampal SWRs.

The most important and novel contributions are in my view: Fig. 4 (distinct hippocampal firing patterns are distinguishable in retrosplenial activity), Fig. 5 (influence of brain state) and Fig. 8 (inhibition of subicular bursting cells prevents SWR propagation to retrosplenial cortex). Overall, this demonstrates that populations activated in retrosplenial SWRs are influenced by the content of hippocampal SWRs and the presence of neocortical (alpha-like) rhythms. This influence is mediated by subicular bursting cells to reach interneurons and pyramidal cells in the superficial layers of the retrosplenial cortex.

Impact

On the one hand, this is a thorough study of the functional pathway during SWR, combining the study of spontaneously occurring SWR with optogenetically induced iHFO, and optogenetic manipulation of specific subicular neuronal types. The characterization of SWR propagation in this circuit is definitively useful for experts investigating the interaction between hippocampus and cortex.

On the other hand, one might regret (likely for technical reasons) that the experimental protocol investigates the SWR phenomenon only during awake immobility, which prevents relating retrosplenial SWR activity to learning or sleep, and thus to clarify the functional role of these events. Two aspects related to functional role of SWR are instead investigated, but with results that are somewhat challenging to interpret (see suggestions below for potential improvements): (1) the interaction of hippocampal SWRs and retrosplenial SWRs, although clearly demonstrating an information transfer between the two structures, does not lead to qualitative differences between hippocampus-coupled and -uncoupled SWRs, although such differences are expected due to the particular role of hippocampus in system consolidation, (2) the modulation of SWR phenomena by neocortical rhythms is also demonstrated, but with limited insights about the underlying mechanisms and function of such modulation.

Suggestions

To maximize the significance of the study, it would be beneficial to characterize in more detail (1) the differences between retrosplenial SWRs occurring or not with hippocampal SWRs, (2) the role of neocortical rhythms.

For (1), one could apply the SWR clustering procedure to retrosplenial SWRs and then characterize

whether there are subtypes preferentially associated to/dissociated from hippocampal SWRs. In addition, coupling with hippocampal SWRs may influence the reliability of the modulation of the retrosplenial rates, which could be investigated based on the same clustering.

For (2), and in line with (1), the paper (as far as I understand) did not clarify whether neocortical rhythms could favor the emergence of retrosplenial SWRs independently from hippocampal input. This question is important as the mechanisms and function of SWRs (apparently) endogenously generated in cortex is largely unaddressed in the literature. To clarify this, one could compare the neocortical state distribution for HP-coupled and HP-uncoupled retrosplenial SWR. In addition, Levenstein et al. (2017) provide interesting insights on how the phase at which action potentials occur within the neocortical rhythms affects plasticity and may lead to a reorganization of the cortical networks. A systematic study of the alpha phase correlates of retrosplenial SWRs (coupled or not with hippocampal activity), may further clarify the functional role of retrosplenial SWRs.

Additional comments:

In the paragraph describing brain state modulation, the statement "In contrast to the unit discrimination of ripple type, which preceded ripple onset (Fig. 4e), LFP magnitude in the 6-12 Hz frequency range prior to SPW-R onset significantly differed according to ripple subtype (Fig. 5g)." is not very convincing: most significant points are located after the ripple onset, and we must keep in mind that wavelet analysis relies on non-causal filters, such that significance maps "leak" towards negative times. In addition, the spot of significant modulation located at (-.6s, 10Hz) may be due to the autocorrelation between successive ripples events (SWRs occurring in pairs could be excluded from the analysis to double check).

References:

Dion Khodagholy, Jennifer N. Gelinias, György Buzsáki (2017) Learning-enhanced coupling between ripple oscillations in association cortices and hippocampus. *Science*.

Daniel Levenstein, Brendon O. Watson, John Rinzel and György Buzsáki (2017) Sleep regulation of the distribution of cortical firing rates. *Current Opinion Neurobiology*.

Michel Besserve

Point to point responses to reviewers' comments on the former manuscript (NCOMMS-19-24417-T):

Reviewer #1 (Remarks to the Author):

Nitzan et al. study the pathway underlying the propagation of ripples from CA1 to cortex. They confirm that the anatomy and connectivity previously described from the CA1 region of the hippocampus to the retrosplenial cortex (via the subiculum) is capable of relaying CA1 ripples to the cortex. Ripples are most likely to be seen in the superficial retrosplenial cortex, but are weaker in the deeper layers. They show that optogenetic activation of CA1 and subiculum can lead to sinks in superficial RSC. Importantly, they show that bursty VGlut2+ neurons in the subiculum are the ones most likely to relay this information. The manuscript represents a substantial amount of solid work and is not far from being ready for publication. It is also impactful and timely, as both the subiculum and retrosplenial cortex are crucial pathways in the flow of information from the hippocampus to the rest of cortex and are rightly receiving a lot of attention right now in the learning and memory field.

The manuscript is appropriate for the journal and will be of interest to the field. However, there are some moderate changes that will help to clarify and improve the manuscript:

1. The authors show that a large proportion of retrosplenial cortex are independent of CA1 ripples (“71% of gRSC ripples did not occur within +/- 25 ms around dorsal hippocampus ripples”). Therefore in Figure 2b, it would be important to include a clear, simple bar graph showing what percentage of RSG-ripples were coincident with CA1 ripples and what proportion were not, in each animal recorded, as well as the average across animals. This is an important point that is made in the text and should be relayed in this figure as well.

Prompted by the reviewer's comment we addressed the coupling between gRSC and hippocampal ripples more carefully. Because our ripple detection algorithm minimized false positives at the expense of false negatives (resulting in undetected events in both areas), we have modified our analysis and terminology: for every ripple event detected in either CA1 or gRSC ripple-band, peak power (within 50 ms window) in both areas was computed and compared (Supplementary Fig. 3h-j showing all ripple events from all wild-type animals). This analysis revealed a much stronger comodulation of ripple power across areas than our previously reported one in which events were compared on a binary basis. Therefore, in the revised version of the manuscript we avoided the 'coupled' or 'uncoupled' terminology, referred to the new quantitative results.

2. In all cases, it appears that “deeper” layers of retrosplenial cortex are also more dorsal because of the steep angles of the probes. This is unavoidable with such medial recordings, but the authors should point out that the “superficial” arrows in most figures also represent “more ventral” locations and discuss in the text how dorsal the “deeper” layer contacts are compared to the “superficial” contacts in each animal and how this could impact interpretation of the data.

We agree that a dorsal-ventral terminology is more appropriate than a deep-superficial one when describing electrode depth. Based on the reviewer's suggestion, we changed the terminology and emphasized in the text that superficial layers correspond to more ventral recordings sites. In addition, we also included in the supplementary histological verification of superficial layer targeting from all animals in which gRSC ripples were described (Suppl. Fig. 1a) and which were not shown in the main figures (Fig. 1a and Fig 2a).

3. The CSD plot in Figure 1D is an important example. In the supplements, the authors should show this for each of the animals.

We now included CSD maps for both gRSC ripples (Supplementary Fig. 1b) as well as hippocampal SPW-R triggered CSD maps (Supplementary Fig. 3). To further demonstrate the coupling of gRSC ripples to negative waves we also include a phase-amplitude coupling plot in the revised version of the manuscript (Supplementary Fig. 1g).

4. Figure 3 and Supplementary Figure 3 use different notations for the intermediate part of the proximodistal CA1 axis. “Intermediate” should be used throughout instead of the more confusing “medial” used in Figure 3.

Done. We have changed the terminology based on the reviewer’s suggestion.

5. On the same topic, the direct (local) response to hippocampal opto stimulation CamKII mice was largest in the proximal area. Given the differences in local response magnitude, the data in Figure 3c regarding RSC responses should be shown both before and after normalization by the local response magnitudes.

To address the reviewer’s point, we now include a new supplementary plot (Supplementary Fig. 5e) and show how the raw gRSC LFP response changes as a function of the stimulation intensity for proximal and distal hippocampal sites. gRSC responses to proximal stimulation remained at baseline levels for all given intensities while responses to distal stimulation increased with increasing intensity.

6. Very few units in RSC increase their firing rates in response to DH stimulation (Fig. 3E). This should be reflected in the Results describing this data and by a simple bar graph clarifying the proportion of neurons excited or inhibited in both superficial and deep layers. The group averages and even the color-coded rasters can make this information hard to discern.

The proportions of the modulated units as well as a pie chart depicting them is now included in the text and figure 3i, respectively.

7. The manuscript focuses on a pathway from CA1 to subiculum to retrosplenial cortex. For this reason using DH (Dorsal Hippocampus) as an umbrella term for CA1 and subiculum is not appropriate and can lead to confusion. The authors need to avoid using DH altogether in this manuscript, and instead use CA1 or subiculum to describe the precise area or sharp-wave ripple they are describing. This is important for the clear interpretation of the results by all readers.

We treated CA1 and subiculum together in the initial part of the manuscript due to our limited number of sessions where CA1, subiculum and superficial gRSC were successfully targeted (n = 4 sessions from 2 animals) and because CA1 and subicular ripples often co-occur (Chrobak and Buzsáki, 1996), rendering the hippocampal reference (CA1/subiculum) for SPW-R evoked responses in gRSC less critical for the interpretation of the cortical response. Based on the comments from reviewers #1 and #2, we now avoided ‘DH’ terminology in the revised version of the manuscript and included a new Supplementary Fig. 4 showing the co-occurrence of ripple in CA1, subiculum and gRSC.

8. Page 2: “The spiking of putative interneurons lagged behind the trough-locked pyramidal neurons by 1 to 2 ms” – in terms of circular variables the “lagged behind” statement is not necessarily true, as there could be a 300 degree phase delay of excitatory neuronal firing during the hyperpolarization induced by the inhibition. This should be clarified.

The delays are now described in angles instead.

9. Additional discussion on the role of VGlut1+ positive cells in the subiculum regarding information encoding and propagation would be valuable given the differences reported here. This should be added to the “Bursting pyramidal neurons...” section in the discussion.

It is interesting to speculate what might be the role of VGlut1+ subicular cells in the propagation of ripple activity to the gRSC. Previous work found that VGlut1+ and VGlut2+ terminals target deep and superficial layers of the RSC in a mutually exclusive manner (Varoqui et al., 2002; REF 67). This finding is confirmed in our study by the different localization pattern of subicular terminals in the gRSC after injection of the Cre-switch or Cre-off vectors

(Supplementary Fig. 9). Given our in vitro observation that the responses of gRSC neurons to stimulation of subicular fibers in the Cre-off experiments were mostly inhibitory, and our in vivo observation that many deep gRSC neurons are inhibited during SPW-Rs, it is likely that VGlut1⁺ projections promote feed forward inhibition of deep pyramidal cells during SPW-R events. However, we should point out that the Cre-off vector used in this study would result in the infection of all subicular (and possibly CA1 to some extent) cells which are non-VGlut2⁺, including local and long-range projecting interneurons (REF. 68), which could potentially mediate such an effect. We therefore feel that the data in our hands is not specific enough to make a firm conclusion about the role of VGlut1⁺ cells in ripple propagation. Nevertheless, we acknowledge the importance of further experiments studying the specific contribution of molecularly-defined subicular subpopulations and examine how each subpopulation supports the hippocampal-cortical dialogue, during ripple as well as non-ripple states and emphasized this point in the Discussion.

Minor change:

1. On page 6: "Likewise, light stimulation of fast spiking interneurons in superficial layers" This makes it sound like the authors are stimulating directly expressing fast spiking neurons. Instead they are stimulating afferent fibers that express opsins while recording from fast spiking neurons in this case. This wording should be revised to make the recording and stimulation configuration clearer.

Thanks. We agree with the reviewer and changed the text to reflect this point.

Reviewer #2 (Remarks to the Author):

This interesting paper reports ripple-like oscillations in RSC layer 2/3 coupled to hippocampal sharp-wave ripples. While the rate of RSC ripples is lower and some are uncoupled to hippocampal events, they correlate with cortical activity packets and negative waves, suggesting embedding dynamics during synchronous 6Hz cortical states. The paper describes the contribution of generic cell-types (PC and interneurons) and look to establish more direct links by manipulating hippocampal and subicular cells with optogenetic tools. Authors suggest that a subset of SUB cells (vglut2+) may play roles in relaying SPW-associated dorsal hippocampal firing to RSC.

The ms is well written, data are overwhelming and results are thought-provoking. I have however a couple of comments regarding specificity that require major clarification.

Major comments:

- Given SPW-r are relatively coherent along the entire hippocampus, it is difficult to establish the specific propagation route. Similarly, given the effect of global brain dynamics and state transitions, it is not surprising to find functional correlation between structures and regions during synchronous brain states. The challenge is to show that some RSC responses associated to dorsal CA1 SPW-ripples are running through or depend on vglut2+ SUB cells. While authors tried to establish the link, the ms reads more like two separate pieces; Fig.1-5 address functional correlations between DH and RSC, while Fig.6-8 look at the effect of manipulating subsets of SUB cells in RSC activity. Data in Fig.7 show activity of optogenetically tagged SUB cells during the local ripple, but not their specificity during RSC ripples coupled/uncoupled to CA1 events. Data in Supp.Fig.8e show that from the total of SPW positively modulated SUB cells (26+58=84) only few were vglut+ as judged by optotagging (26/84= 30%). The ms will strongly benefit from reinforcing specificity of vglut2+ SUB cells in coupling hippocampal activity to RSC during some SPW ripples. I guess authors can exploit their data to stress this point further.

We would like to emphasize that the opto-tagged subicular cells likely represent an underestimation of the true VGlut2⁺ population due to multiple factors such as viral transfection efficacy, reduced activation on neighboring non-illuminated shanks and the threshold for classifying a unit as light-responsive. As a comparison, using similar methods in Senzai et al (Neuron, 2017) only 27% (29/107) of the physiologically identified putative mossy cells were responsive to light stimulation when using a cell-type specific Cre line. Given our observation that activation of VGlut2⁺ Chr2-expressing subicular cells led to an increase in ripple band power in superficial layers of the gRSC, which is now further supported by using additional subjects and analyses (see below), and in the absence of other known subpopulations of subicular projection neurons that target superficial layers of the gRSC (see REFs 67-70) we believe that we have established a link between the activity of VGlut2⁺ subicular and gRSC ripples. For the experiments described in Fig. 7, recordings were performed only in the subiculum and gRSC activity was not monitored. We repeated the procedure of identifying light-responsive pyramidal units in the additional animals used for Fig. 8 and show that the activity of the majority of these units is up-modulated during spontaneous gRSC ripple events and precedes the onset in activity of gRSC neurons (Supplementary Fig. 11d).

- Similarly, specificity of optogenetic evidence should be more carefully addressed. First, there is mismatch between the RSC LFP response after CA1 stimulation (Fig.3b,d) versus vglut2+ SUB stimulation (Fig. 8c). While ripple-like oscillations are evoked all along the 100 ms light pulse in CA1, only an early wave is evoked in response to SUB stimulation. It is unclear whether the high frequency spectrum shown in Fig.8d reflects more spiky wave components than a real LFP ripple oscillation. Second, RSC unit responses differ in CA1 (Fig.3e) versus vglut SUB stimulation (supp.fig.9a) in terms of rate, tonic firing dynamics and rebounds. Finally, the dynamics of SUB induced packet rates (Fig.8e) does not match with that of spontaneous ripples (Fig.5d) suggesting there is an additional delay when SUB cells are directly activated. It is therefore unclear whether optogenetic induced activity at CA1 and SUB is targeting RSC through different pathways.

We thank the reviewer for bringing up this issue. However, there might be - in part - a small misunderstanding due to our lack of clarity of explanation. When quantifying the differences between optogenetic induced responses in CaMKII-Cre::Ai32 and VGlut2-cre animals (both of the LFP and unit activity), it is important to keep in mind the different Chr2 delivery strategies used in each animal model. While every single CaMKII-expressing (putative excitatory) neuron in CaMKII-Cre::Ai32 animals is expected to express Chr2, for Chr2 delivery in VGlut2-cre mice we used viral injections of highly conservative amount (100 nL, titer $1.6 \cdot 10^{12}$ VG/mL) in order to avoid viral spread into the gRSC, where VGlut2 is also sparsely expressed. As a result, the available pool of Chr2-expressing subicular neurons that can potentially mediate light evoked responses in the gRSC in VGlut2 mice is expected to be much smaller than that in CaMKII-Cre::Ai32 mice. In addition, the two sets of experiments employed different light delivery protocols: in CaMKII-Cre::Ai32 animals we used a square pulse stimulation for maximal light delivery allowing to uncover weak connections and have a define onset. In contrast, stimulation in VGlut2-Cre animals consisted of both square pulses (allowing for connectivity mapping due to the higher amount of light delivered and the well-defined start/end point of 'effective' light delivery) as well as half sine waves to avoid the edge artifacts caused by the square pulses, thereby favoring the induction of physiologically-resembling ripple events. To ease the interpretation of the results we now indicate what pulse shape was used in every experiment.

To address the reviewer's second point regarding the reliability of the stimulation-triggered spectrograms, we took several extra measures that support our initial findings. First, we increased our sample size and statistical power by repeating the experiments in two additional subjects. Second, we identified cortical ripple events independently of the stimulation and computed the cross-correlogram between the stimulations and gRSC ripples (Supplementary Fig. 11c), which shows a peak in gRSC ripple rate following the onset of light stimulation. Third, we include exemplary LFP traces of gRSC ripples induced by subicular stimulation (Fig. 8c). Forth, we increased the range of frequencies shown in the spectrograms up to 250 Hz to demonstrate that the observed increase in spectral power is confined to the ripple band and does not 'bleed' into higher frequencies as would be expected from spiky wave components (cf. Fig. 5a middle panel).

As for the different dynamics of ripple-packets and stimulation packets coupling, the most parsimonious explanation is that our stimulation induced more inhibition compared with spontaneous ripples. This is suggested by the larger proportion of units inhibited by stimulation as compared to ripples (cf. Fig.2h and Supplementary Fig. 11a). The increase in inhibitory tone may result in a transient silence period, whereas the observed increase in packet rate reflects a synchronous rebound spiking. Furthermore, the mechanisms underlying the coupling between hippocampal ripples and cortical packets (or UP and DOWN states) are not well understood and it is likely that other, third party unrecorded areas contribute as well. Our manipulations were restricted to a single partner (the subiculum) of the hippocampal-cortical dialogue and they do not offer mechanistical insights about the nature of this coupling. They do suggest, however, that VGlut2⁺ subicular cells likely contribute to this coupling as suggested by computational modeling where the occurrence of SWP-Rs temporarily disrupts the cortical excitatory/ inhibitory balance (Levenstein, Buzsáki and Rinzel 2019).

Other comments:

- In their analysis, data from CA1 pyramidal layers and SUB are pooled collectively into DH (unclear contribution), but these regions are different in terms of PC identity and connectivity with RSC (thus part of heterogeneity may be region-specific). Authors may reinforce direct links between the dorsal CA1 and RSC “via” SUB by identifying CA1, SUB and RSC responses separately. This will help addressing my first major point.

Done. Please see also response to point #7 of reviewer #1.

- Authors use a burstiness index to evaluate the firing autocorrelogram in vivo and to presumably establish links with in vitro data. SUB bursting cells are defined by their response to current injection in vitro. Evaluation of complex spike behavior by the firing autocorrelogram is not directly related with the intrinsic bursting phenotype. Complex spikes are dendritically generated and obey to different mechanisms. For instance, CA1 pyramidal cells fire regularly when tested in vitro but exhibit complex spikes in vivo. This should be clarified to avoid confusion.

We agree with the reviewer that the burst firing seen in whole-cell recordings and defined by the fused waveform of successive action potentials and complex spikes observed in extracellular recordings may not represent exactly the same phenomena. Nevertheless, recent evidence suggests that burst propensity seen in-vitro often relate to complex spike emission seen extracellularly (see for example Hunt et al., Nature Neuroscience 2019, for the case of CA3 pyramidal cells). Moreover, bursting and non-bursting subicular cells have also been described in extracellular unit recordings (for review see O’mara, Behav. Brain Research 2006). Based on the reviewer’s criticism, we have now change the wording to reflect this point of ambiguity and to avoid confusion.

- Difference in burstiness of RSC-ACC cells and their potential association with hippocampal ripples was previously reported by Wang and Ikemoto (ref 17).

Due to the relatively small sample size of superficial gRSC units and the large heterogeneity in burstiness among deep pyramidal cells, both in vivo and in vitro, we did not compare quantitatively the burstiness of ripple modulated and unmodulated units.

- Hippocampal iHFO looks more like pathological population spikes than physiological ripples (the amplitude, firing rate, spectral leakage of iHFO events in Supp.Fig 3b) suggest that high intensity optogenetic stimulation of CA1 in CaMKII animals is eliciting hypersync responses. Can more physiological events induced with lower light intensity? Please, clarify this point.

As previously demonstrated by Stark et al (2014), iHFO power and frequency increase with light intensity, and, with the appropriate selection of stimulation intensity and waveform spontaneously-looking ripple events can be induced (Stark et al., 2014). For the mapping experiments in the present study (Fig. 3), we chose to use a high-intensity square pulse stimulation to drive the post synaptic cells beyond their physiological magnitude

during natural ripple events and to uncover weak connections not seen otherwise. Using optogenetic stimulation in VGlut2-Cre animals, we also demonstrate that oscillatory events mimicking spontaneous ripples can be elicited with lower intensity of stimulation.

- While authors acknowledge the potential confounding effect of direct GABAergic CA1 projections to RSC (ref 67) their real impact in spontaneous and induced RSC oscillations is not addressed (e.g. by some pharmacological experiment). Similarly, optogenetic stimulation has direct and indirect microcircuit effects which can explain part of heterogeneity and unspecific responses. Any additional experiment or analysis addressing these issues will make the paper stronger.

As the reviewer mentioned, we emphasize the potential role of long-range GABAergic projections from CA1 area to the gRSC in cross-regional synchronization of ripple events and as a potential additional source of ripple-associated inhibition seen mainly in deep gRSC pyramids (together with local inhibition by gRSC interneurons). We have performed an initial set of experiments in which we attempted to label retrogradely long-range interneurons from CA1 to gRSC. However, we did not see many interneurons in CA1 projecting to the gRSC. However, in the absence of specific molecular markers we cannot address their impact in this study. Therefore, we feel unable to adequately address this point.

- Regarding optogenetic tagging of vglut2+ units inhibited by light stimulation reflect microcircuit effects; it is not necessarily indicative of a vglut2- phenotype.

We agree and attempted to make this point clear in the manuscript.

- Arch experiments: why was a 50 ms delay chosen? While there is a significant reduction of RSC ripple reported in Fig.8g, raw data in Fig.8f raise doubts on how was light stimulation triggered and evaluated. First, the hippocampal ripple preceding light occurred more than 50 ms before, so it is unclear how is delay defined. Second, the ripple power is very low as compared with those following so it is unclear how was closed-loop defined. Third, while the power of the RSC ripple-like events is reduced by green light, multi-unit activity survives and some units are actually activated (possibly by disinhibition) as shown in Fig.8h. All this suggest complex microcircuit effects underlying hippocampal communication.

The closed loop was defined by filtering one hippocampal channel in the ripple band, on which the laser was triggered by crossing of a manually set threshold. As the reviewer pointed, in many cases the delay to stimulation was more than 50 ms. This may be both due to variations in the exact time point of threshold crossing, or because of technical limitation of our laser system. To address this issue, we now computed the delay to stimulation for every detected event and report this value instead. In addition, we replaced Fig. 8f with multiple different examples. We also discuss the point raised by the reviewer about complex circuit effects caused by the stimulation. However, we should also point out that the persistence of multi-unit activity during stimulation is expected since we inhibit only the subicular fiber terminals, while gRSC cells receive other inputs as well.

- Remondes and Wilson reported local RSC gamma oscillations coupled to hippocampal SPW ripples (though at more rostral levels). This is somehow suggested by spectrograms in Fig.5 and low frequency components in Fig.2g. It is also suggested by the modulation of PETHs shown in Fig.2h. Please, address and clarify.

The increase in cortical gamma power during ripple events likely reflects the coupling of hippocampal SPW-Rs to cortical activity packets which are accompanied by an increase in gamma, as well as high frequencies power (Fig. 5a). This point is now discussed under 'Embedding of hippocampal SPW-Rs in cortical rhythms'.

- page 7, section Identification of SUB "... striking differences in burstiness (Fig.5b bottom, 5d...)" should be Fig.7b,d.

Thus inter regional coordination may be running through gamma and not only during ripples. Please, specify panel numbers when referring to Supp.Fig.8 in this section.

Done.

- terminology is not homogenous along the ms, which complicates reading. For instance, in sup.fig8e positively or negatively modulated refer to SPW and responsive/not-responsive refers to light, while in supp.fig.9b modulation refers to light.

In Fig. 7 and Supplementary Fig. 10 we used a binary classification (up-modulated or not) for light-evoked responses to identify light-responsive subicular units. This point is made clear in the main text: "Only units that displayed a significant increase in firing rate within the 10-ms long light pulse were considered directly light responsive". In contrast, in Supplementary Fig. 11 we show that light stimulation can result in both up- or down regulation of firing rates in gRSC and hence positively or negatively modulated.

- page 2, CSD map (Fig.1d) not e. Also, the probe shank is not parallel to the somatodendritic axis which may complicate interpretation of CSD. This is alleviated by using ICA, but worth noting.

This point is now emphasized according to the reviewer's suggestion. In addition, gRSC ripple triggered and hippocampal SPW-R triggered CSD maps for all individual subjects are now added to the supplementary.

- Sup.Fig.2 is called before Sup.Fig.1

Fixed.

- Methods should be more detailed. For instance, event detection (SPW, ripples, negative waves, packets) is dependent on a threshold which should be more carefully addressed. Event clustering is implemented by k-means with k=10 without justification.

A justification for k=10 in clustering analysis is now shown in Supplementary Fig. 6c.

- Clustering SPW analysis is very interesting (Fig.4). However, statistical significance of cluster specificity should be challenged with some additional methods and incorporated into the figure. Some rasters appear only marginally significant.

To address the reviewer's point, we used a t-SNE clustering algorithm, which supports our initial findings regarding the existence of multiple types of hippocampal SPW-Rs (Fig. 4 and Supplementary Fig. 6).

- Interneurons contribute heterogeneously to ripples in both hippocampal and cortical regions (Klausberger & Somogyi; Averkin & Tamas). While criteria for unit separation do not allow to disambiguate (see heterogenous contribution in Fig. 1g, 2i,j) it may be important to make a note on this point.

Indeed, because of the lack of interneuron-specific marker lines, any statements regarding the contribution of specific interneurons subpopulations should not be made here. However, as our data clearly suggest, there exist a large degree of heterogeneity in SPW-R evoked responses among cortical interneurons. This point is now emphasized in the text.

- Some terms and analyses are not standard and the general reader may have difficulties. The narrative will improve by including short descriptions to facilitate interpretation. For instance, cross-covariance.

A short description of cross-covariance analysis is now added to the main text. In addition, a full description of this analytical procedure, including the formula, is included under materials and methods.

- Fig.3: Distal optogenetic stimulation of CA1 may work better than intermediate or proximal stimulation due to off-target direct illumination of RSC terminals, especially because strong intensity was used. Can authors discard such an effect?

We can rule out the off-target direct illumination of RSC cells for two reasons: first, hippocampal responses decayed to baseline within a distance smaller than that to the nearest RSC recording site. This is now included as Supplementary Fig. 5f. Second, the hippocampal probe, and its mounted fiber optics or μ LEDs was inserted such that CA1 probe was in between the light source and the gRSC probe, which was implanted at a 15° dorsal/ventral angle in the medial to lateral orientation, thus rendering the spread of light 'backwards' in gRSC direction unlikely.

- check spelling and some typos (e.g. electromyographic..)

Done.

- Fig. 7b, please add labels to facilitate reading. Also, caption identifies blue cells as bursting and red as regular-spiking cells without a clear criterion.

Done.

Reviewer #3 (Remarks to the Author):

This manuscript addresses the propagation of sharp wave-ripple (SWR) complexes to the retrosplenial cortex. A combination of In vivo (including optogenetics manipulations) and slice experiments are used to characterize the functional pathway from the hippocampus to the retrosplenial cortex during SWR in mice.

SWR are considered instrumental for the two stages model of memory trace formation as retrieval, which both require the broadcast of these highly synchronous events to cortex (e.g. to induce long-term potentiation in cortico-cortical connections). Evidence for the relationship of SWR to cortical activity can be found in the literature, and it has been noticed in particular that hippocampal SWR coincide with cortical SWR in several associative areas, including the retrosplenial cortex (Khodagholy et al. 2017). Besides this study, SWR in the retrosplenial cortex have not been investigated, as far as I know, although the contribution of this region to memory functions has been supported by multiple studies.

Content of the study

The experimental result demonstrate a very good command of electrophysiology and optogenetic techniques leading to high quality data. The data analysis is well executed. The study reproduces many aspects of the previous literature, witnessing in particular the occurrence of cortical SWRs with clear electrophysiological properties, both at the level of the LFP and spiking activity (Fig. 1). As also previously reported, a fraction of these SWRs co-occur with hippocampal SWRs.

The most important and novel contributions are in my view: Fig. 4 (distinct hippocampal firing patterns are distinguishable in retrosplenial activity), Fig. 5 (influence of brain state) and Fig. 8 (inhibition of subicular bursting cells prevents SWR propagation to retrosplenial cortex). Overall, this demonstrates that populations activated in retrosplenial SWRs are influenced by the content of hippocampal SWRs and the presence of neocortical (alpha-like) rhythms. This influence is mediated by subicular bursting cells to reach interneurons and pyramidal cells in the superficial layers of the retrosplenial cortex.

Impact

On the one hand, this is a thorough study of the functional pathway during SWR, combining the study of spontaneously occurring SWR with optogenetically induced iHFO, and optogenetic manipulation of specific subicular neuronal types. The characterization of SWR propagation in this circuit is definitively useful for experts investigating the interaction between hippocampus and cortex.

On the other hand, one might regret (likely for technical reasons) that the experimental protocol investigates the SWR phenomenon only during awake immobility, which prevents relating retrosplenial SWR activity to learning or sleep, and thus to clarify the functional role of these events. Two aspects related to functional role of SWR are instead investigated, but with results that are somewhat challenging to interpret (see suggestions below for potential improvements): (1) the interaction of hippocampal SWRs and retrosplenial SWRs, although clearly demonstrating an information transfer between the two structures, does not lead to qualitative differences between hippocampus-coupled and -uncoupled SWRs, although such differences are expected due to the particular role of hippocampus in system consolidation, (2) the modulation of SWR phenomena by neocortical rhythms is also demonstrated, but with limited insights about the underlying mechanisms and function of such modulation.

Suggestions:

To maximize the significance of the study, it would be beneficial to characterize in more detail (1) the differences between retrosplenial SWRs occurring or not with hippocampal SWRs, (2) the role of neocortical rhythms.

For (1), one could apply the SWR clustering procedure to retrosplenial SWRs and then characterize whether there are subtypes preferentially associated to/dissociated from hippocampal SWRs. In addition, coupling with hippocampal SWRs may influence the reliability of the modulation of the retrosplenial rates, which could be investigated based on the same clustering.

Please see response to reviewer #1 point #1 regarding ripple coupling. Due to the much smaller number of gRSC ripples per session compared with CA1 ripples applying the same clustering procedure to gRSC ripples was not feasible.

For (2), and in line with (1), the paper (as far as I understand) did not clarify whether neocortical rhythms could favor the emergence of retrosplenial SWRs independently from hippocampal input. This question is important as the mechanisms and function of SWRs (apparently) endogenously generated in cortex is largely unaddressed in the literature. To clarify this, one could compare the neocortical state distribution for HP-coupled and HP-uncoupled retrosplenial SWR. In addition, Levenstein et al. (2017) provide interesting insights on how the phase at which action potentials occur within the neocortical rhythms affects plasticity and may lead to a reorganization of the cortical networks. A systematic study of the alpha phase correlates of retrosplenial SWRs (coupled or not with hippocampal activity), may further clarify the functional role of retrosplenial SWRs.

As discussed in the response to reviewer #1 point #1, based on our new cross-regional power correlational analysis we avoided the binary classification of ripples in the revised manuscript. However, given the reviewer's suggestion, we examined the phase modulation of gRSC units by 6-12 Hz cortical negative waves using our previous binary classification for coupled and uncoupled events. We did not find differences in either preferred phase or coupling strength between the two groups. These results may again reflect undetected hippocampal ripple events, or they may imply that the phase locking to negative waves reflecting synaptic inputs are independent on the source of that input.

Phase coupling of significantly modulated gRSC units ($n = 106$) to negative waves coupled (top) or uncoupled (bottom) with CA1 ripples. Mean population ITPC (radial axis, grey) for coupled events: 0.40 ± 0.01 ; uncoupled events: 0.38 ± 0.01 . Mean preferred angle coupled events: 199.15 ± 4.45 ; uncoupled events: 195.19 ± 4.10

Additional comments:

In the paragraph describing brain state modulation, the statement “In contrast to the unit discrimination of ripple type, which proceeded ripple onset (Fig. 4e), LFP magnitude in the 6-12 Hz frequency range prior to SPW-R onset significantly differed according to ripple subtype (Fig. 5g).” is not very convincing: most significant points are located after the ripple onset, and we must keep in mind that wavelet analysis relies on non-causal filters, such that significance maps “leak” towards negative times. In addition, the spot of significant modulation located at (-.6s, 10Hz) may be due to the autocorrelation between successive ripples events (SWRs occurring in pairs could be excluded from the analysis to double check.

We accept the reviewer’s criticism that wavelet analysis can result in spectral leakage around the center of the wavelet. However, the extent of this spectral leakage at each frequency is proportional to that frequency and depends on the standard deviation of the Gaussian which was used to generate the wavelet. A leakage of 0.5-1 seconds for 6-12 Hz (were most significant pre-ripple points are observed) is therefore not likely (using our parameters for a 6 Hz wavelet non-zero points of the wavelet extended maximally to -300 ms). In contrast, pre-ripple significant points around 2 Hz in Fig. 5g may indeed reflect spectral leakage (and were therefore not discussed in the manuscript). Regarding the reviewer’s second point, ripple bursts were excluded from the analysis (throughout the entire manuscript, only events with inter-ripple intervals > 0.5 s were taken).

Reviewers' Comments:

Reviewer #1:

Remarks to the Author:

The authors have addressed many of my concerns and comments. However two major issues remain:

1) The authors are currently making two somewhat conflicting statements. On the one hand, only a fraction of retrosplenial ripples are modulated by hippocampal ripples using a discrete thresholded (what the authors call "binary") analysis. On the other hand power over 50 ms windows is correlated, as shown in the supplementary figure. This is obscure and unclear. Both methods should be shown in the same (main) figure and summarized and a cohesive and clear explanation provided for these statements - one that is clear from the figures presented. This is one of the central points of this paper, and clarity on the central point is critical here. Are the majority of retrosplenial ripples triggered by hippocampal ripple activity or not? The answer is central to determining the proposed role of information transfer from the hippocampus to the cortex via the subiculum->retrosplenial pathway.

2) The authors' central result replicates work from Dong Wang's lab, as such needs to be put in context of those results (Opalka et al., "Hippocampal Ripple Coordinates Retrosplenial Inhibitory Neurons during Slow-Wave Sleep"). Opalka et al show that hippocampal ripples are associated with activation of inhibitory putative fast-spiking neurons and what appears to be strong suppression of putative excitatory neurons in retrosplenial cortex. Comparing Figure 3 of the Opalka et al paper to Figure 2 of the current manuscript shows that many findings are overlapping. However, the firing rate of putative excitatory neurons seems to show somewhat different results. The authors should acknowledge the Opalka paper and also compare their results in detail to those reported in that paper.

Reviewer #2:

Remarks to the Author:

Authors have addressed some of my comments. I found however that most revisions are incorporated into supplementary figures but not in principal figures. This should be corrected because many of the previous concerns still arise. In particular:

- I still have concerns regarding specificity on the propagation route as claimed by authors (my first major point). I specifically asked for data supporting that at least some RSC responses associated to dorsal CA1 SPW-ripples are running through or depend on vglut2+ SUB cells. Data in Fig.7 show activity of optogenetically tagged SUB cells during the local ripple, but not their specificity during RSC ripples coupled/uncoupled to CA1 events. Authors now have added data on optotagged SUB cells in response to RSC ripples (Supp.Fig.11). I found these data should be added to Fig.7.

- My point on specificity of optogenetic experiments still arises. Authors have addressed some of my concerns (in response to my second point) but I am still unclear why a delay is implemented for closed-loop experiments, which are critical to prove causality. Intuitively, if you want to test specificity of CA1-SUB-RSC event transmission you typically think on detecting the CA1 event by closed-loop, tightly inhibiting SUB cells and confirming no event propagated to RSC. Authors have improved Fig.8 with new data and analysis, but I still do not see the reason why not to do the direct experiment (no delay). Arguing these are states with high ripple occurrence is not guarantee of specificity per se. The ms will benefit for running this experiment specifically.

- Together with R1 we raised concerns regarding how authors analyzed and identify coupling between

CA1-SUB and RSC ripples early in Fig.2. Authors have addressed this in a subset of 4 sessions from 2 mice and data reported in Supp.Fig.4. This should be added to Fig.2. Similarly, data in Supp.Fig.3h (for CA1 to RSC) should be displayed in the principal figure and a similar plot for SUB-RSC data should be provided. Also, correlations should be validated statistically.

- I still have concerns on the physiological nature of some optogenetically induced ripples. Authors support on Stark et al., but that paper also raises concerns regarding the same issue. As they acknowledge the power of iHFO can be parametrically controlled meaning that depending on expression level and network effects the activity may be pathologically high. Thus, some of the experiments are not really testing the physiological route CA1-SUB-RSC. That is OK for confirming the target region (though it will be still confounded by downstream micircruit amplification) but it is not specific of ripple events. Similarly, as authors recognize induced events may recruit more inhibition than physiological events. Therefore, spontaneous CA1-SUB-RSC events and induced events may actually link to different circuitry. All this should be reflected and considered in the ms.

- A recent paper by Remondes lab (Ferreira-Fernandes et al., Cell Reports 2019) supports direct connectivity from CA1 to RSC, not necessarily through the SUB. I still feel authors need to include this view.

- I appreciate clarification on k-means and including t-SNE clustering. Still feel this analysis is too canned. For instyanmce k-means may have issues regarding initialization resulting in different clusters for each realization. Similarly, t-SNE may varies a lot depending on the seed used, learning rate and cost function used. None of this is really explained. Please, clarify methodological effects.

Reviewer #3:

Remarks to the Author:

If find the authors' rebuttal satisfactory, and in particular I am happy that this revised version clarified the coupling between hippocampal and cortical ripples.

The results are novel and provide important insights regarding the communication between hippocampus and neocortex.

Reviewers' comments:

Reviewer #1 (Remarks to the Author):

The authors have addressed many of my concerns and comments. However two major issues remain:

1) The authors are currently making two somewhat conflicting statements. On the one hand, only a fraction of retrosplenial ripples are modulated by hippocampal ripples using a discrete thresholded (what the authors call "binary") analysis. On the other hand power over 50 ms windows is correlated, as shown in the supplementary figure. This is obscure and unclear. Both methods should be shown in the same (main) figure and summarized and a cohesive and clear explanation provided for these statements - one that is clear from the figures presented. This is one of the central points of this paper, and clarity on the central point is critical here. Are the majority of retrosplenial ripples triggered by hippocampal ripple activity or not? The answer is central to determining the proposed role of information transfer from the hippocampus to the cortex via the subiculum->retrosplenial pathway.

We thank the reviewer for his/her suggestion. The event-wise cross-correlation and power-power cross-correlations offer different, but complementary insights. Cross-correlation of discrete events can provide information about the temporal delays between two events. In this case it suggests that the majority of gRSC ripples follow CA1 ripples as indicated by peak at positive values in Fig. 2d. However, as we discussed in our previous response letter, this analysis cannot account for false negatives (undetected events). Our power-power correlation shows that there is a positive correlation between CA1 ripple and gRSC ripple power implying that stronger hippocampal ripples are more prone to propagate to the gRSC. It also shows that the majority of gRSC ripples are associated with significant (above detection threshold) hippocampal ripple-band power, while only a fraction at the lower left bottom of the plot is not. We welcome the reviewers' comment that both plots should have been shown in the same figure and now moved those panels to main figures. We present them as density plots, which is more appropriate to illustrate the existence of two separate groups of gRSC ripples (new Fig. 2e), together with probability density plots of simultaneously recorded CA1, SUB and gRSC ripples. Because with these additions the figure would be excessively busy, we split Fig. 2 into two figures.

2) The authors' central result replicates work from Dong Wang's lab, as such needs to be put in context of those results (Opalka et al., "Hippocampal Ripple Coordinates Retrosplenial Inhibitory Neurons during Slow-Wave Sleep"). Opalka et al show that hippocampal ripples are associated with activation of inhibitory putative fast-spiking neurons and what appears to be strong suppression of putative excitatory neurons in retrosplenial cortex. Comparing Figure 3 of the Opalka et al paper to Figure 2 of the current manuscript shows that many findings are overlapping. However, the firing rate of putative excitatory neurons seems to show somewhat different results. The authors should acknowledge the Opalka paper and also compare their results in detail to those reported in that paper.

We thank the reviewer for pointing us to this paper, which was published after the submission of our revised manuscript. A close examination of the results presented in Opalka et al. indicates that,

under their experimental settings (deep cortical layers recordings in naturally sleeping rats), CA1 SPW-Rs tended to occur at the transition from an UP to a DOWN state, rather than from a DOWN to an UP state (or at the onset of an activity packet as we observed). Since DOWN-states reflect global cessation of firing it is not surprising that the authors observed a net inhibitory effect of SPW-Rs on cortical cells. Moreover, as cortical interneurons tend to increase their activity toward the end of the UP-state (Zucca et al., Elife 2017), one would also expect to observe an increase in firing among interneurons around the time of SPW-Rs, as reported by Opalka and colleagues. Altogether, it is more likely that the observations made by Opalka et al. in deep cortical pyramids are not directly driven by hippocampal outputs, but are governed by cortical dynamics (i.e. UP and DOWN states) which, in turn, are coupled to hippocampal SPW-Rs. The increased firing of gRSC interneurons around SPW-Rs is an exception and may reflect a combination of direct SPW-R – associated excitation (as our in-vitro data directly shows - both superficial and deep layers interneurons receive monosynaptic inputs from the hippocampus), and intrinsic UP-state related cortical dynamics.

It is noteworthy that the observations made by Opalka et al., namely a coupling of SPW-Rs to the UP->DOWN transition, are only partially in agreement with other studies (e.g. Peyrache et al., 2009) and are contradictory to previous results from the retrosplenial cortex (Battaglia et al., 2004) and other cortical areas (Wilber et al., 2017; Khodagholy et al., 2017; Isomura et al., 2006), where global cortical activity (mostly contributed by L5 pyramidal neurons) is increased around SPW-Rs. The reasons for this variability in coupling dynamics are not entirely understood and are discussed in Levenstein et al., 2019. Therefore, interpretation of unit responses in one area to oscillatory events in another structure should be done after considering (i) known synaptic connectivity and (ii) local currents generators. We tried to emphasize this point in the original version of our manuscript and expanded this line of arguments now, including a citation of the new study.

Reviewer #2 (Remarks to the Author):

Authors have addressed some of my comments. I found however that most revisions are incorporated into supplementary figures but not in principal figures. This should be corrected because many of the previous concerns still arise. In particular:

- I still have concerns regarding specificity on the propagation route as claimed by authors (my first major point). I specifically asked for data supporting that at least some RSC responses associated to dorsal CA1 SPW-ripples are running through or depend on vglut2+ SUB cells. Data in Fig.7 show activity of optogenetically tagged SUB cells during the local ripple, but not their specificity during RSC ripples coupled/uncoupled to CA1 events. Authors now have added data on optotagged SUB cells in response to RSC ripples (Supp.Fig.11). I found these data should be added to Fig.7.

We appreciate the reviewer's suggestion. However, we do not see how Supp. Fig. 11d would relate to Fig. 7 which is entitled "Identification of subicular bursting cells in vivo" and aims to show the specificity of the VGlut-cre marker in labeling bursting subicular cells. The specificity of VGlut2⁺ subicular cells in evoking ripple activity in the gRSC is the topic of Fig. 8 (now Fig. 9), where we directly demonstrate that prolonged optogenetic stimulation of those cells results in gRSC ripples. We assume that the reviewer intended to refer to Fig. 8. Based on the reviewer's previous comments, we have corroborated these findings with new data and analysis including both raw data and event-wise cross-

correlation. The demonstration that subicular bursting cells activity is increased during gRSC ripples is correlational, not causal. Based on these considerations, we hope that the reviewer agrees that it is more appropriate to present these findings in a supplementary figure to Fig. 9.

- My point on specificity of optogenetic experiments still arises. Authors have addressed some of my concerns (in response to my second point) but I am still unclear why a delay is implemented for closed-loop experiments, which are critical to prove causality. Intuitively, if you want to test specificity of CA1-SUB-RSC event transmission you typically think on detecting the CA1 event by closed-loop, tightly inhibiting SUB cells and confirming no event propagated to RSC. Authors have improved Fig.8 with new data and analysis, but I still do not see the reason why not to do the direct experiment (no delay). Arguing these are states with high ripple occurrence is not guarantee of specificity per se. The ms will benefit for running this experiment specifically.

As noted in our previous response letter, the delay in stimulation was not deliberately implemented, but reflects a technical limitation of our laser system. Therefore, in our current settings running the 'direct experiment' with no delay is not feasible. Importantly, the delay issue will have no impact on our main conclusions.

We chose to use a relatively long period of stimulation which allowed us to 'capture' sufficient numbers of hippocampal SPW-Rs necessary for proper statistical comparison. Moreover, some evidence suggests that the Arch-mediated terminal inhibition is partially mediated by the extracellular acidification which requires longer stimulation period to take effect (El-Gabi et al., 2016). While we agree that all new experiments would add value to our manuscript, the experiments suggested by the reviewer would require to set up transgenic animal breeding, perform injections and wait 6-8 weeks for the expression of the virus before we can actually perform the experiments and analyze the data. In the initial round of revision, this comment was deemed as 'minor'. Therefore, we hope that the reviewer agrees that the main message of our manuscript would not be compromised by the absence of the suggested experiment.

- Together with R1 we raised concerns regarding how authors analyzed and identify coupling between CA1-SUB and RSC ripples early in Fig.2. Authors have addressed this in a subset of 4 sessions from 2 mice and data reported in Supp.Fig.4. This should be added to Fig.2. Similarly, data in Supp.Fig.3h (for CA1 to RSC) should be displayed in the principal figure and a similar plot for SUB-RSC data should be provided. Also, correlations should be validated statistically.

Please see response to reviewer #1 point #1.

- I still have concerns on the physiological nature of some optogenetically induced ripples. Authors support on Stark et al., but that paper also raises concerns regarding the same issue. As they acknowledge the power of iHFO can be parametrically controlled meaning that depending on expression level and network effects the activity may be pathologically high. Thus, some of the experiments are not really testing the physiological route CA1-SUB-RSC. That is OK for confirming the target region (though it will be still confounded by downstream micrcircuit amplification) but it is not specific of ripple events. Similarly, as authors recognize induced events may recruit more inhibition than physiological events.

Therefore, spontaneous CA1-SUB-RSC events and induced events may actually link to different circuitry. All this should be reflected and considered in the ms.

Perturbation experiments are typically designed to either mimic physiological patterns or emphasize certain aspects of those patterns. iHFOs can capture multiple aspects of native ripples, including their oscillation frequency range, duration and the temporal relationship among pyramidal neurons and interneurons. Despite variations in opsin expression levels and light scattering, the firing rates of individual pyramidal cells and interneurons during ripples and iHFOs strongly correlate. Even the sequential firing of neurons during ripples and iHFOs show a reliable correlation (Stark et al., PNAS 2015). On the other hand, increasing light intensity can transform physiology-mimicking iHFOs to exaggerated patterns with a large fraction of pyramidal cells firing synchronously. These aspects of iHFOs have been emphasized in all previous papers as well as in our manuscript. In Fig. 3 (now 4), we used relatively high intensity levels to probe and map the topology between the structures. Here we do not claim that these patterns are physiological. This was a prerequisite for the hypothesis that physiological ripples can take the CA1-SUB-RSC route (Fig.4 g-j show that the direction of the cortical response to opto stim correlates with the timing of natural ripples). Moreover, these experiments do corroborate our in-vitro findings and show a deep-superficial gradient of responses. They also indicate an anterior-posterior gradient in responses along the RSC which is in-line with the gradients in density of subicular fibers in the RSC (Supplementary Fig. 8). Therefore, we find that these experiments are useful in confirming connectivity in vivo. For a more physiological replication of the gRSC correlates of spontaneous ripples, light power in the experiments for Fig. 9 was adjusted such that spontaneous and induced ripples were practically indistinguishable (e.g., Fig. 9a). Yet, we caution, as advised by the reviewer, that perturbation experiments should always go hand in hand with correlational data of unperturbed circuits (discussed under “Bursting pyramidal neurons in dorsal subiculum convey hippocampal SPW-R messages to gRSC”).

- A recent paper by Remondes lab (Ferreira-Fernandes et al., Cell Reports 2019) supports direct connectivity from CA1 to RSC, not necessarily through the SUB. I still feel authors need to include this view.

We thank the reviewer for pointing us to this paper, which was published after the submission of our revised manuscript. Indeed, sparse projections from CA1 area could explain why we observed residual EPSPs in our Cre-off in-vitro experiments. This point is now added to the manuscript under the respective section.

- I appreciate clarification on k-means and including t-SNE clustering. Still feel this analysis is too canned. For instance k-means may have issues regarding initialization resulting in different clusters for each realization. Similarly, t-SNE may vary a lot depending on the seed used, learning rate and cost function used. None of this is really explained. Please, clarify methodological effects.

We thank the reviewer for pointing out the omission of some technical details about algorithmic initialization, which are now added to the Methods section. The goal of the t-SNE analysis was to show that neural states during ripples (as defined by the pattern of spiking activity) in CA1 and gRSC do covary. Clustering is a means by which to parse this variance. A clustering approach was motivated by the observation that cluster discrimination saturates at around N = 10 clusters and that neural spiking does

not uniformly tile representational space, as shown by our dimensionality reduction. K-means is a standard first-pass algorithm, and without knowing how post-synaptic neurons discriminate patterns of presynaptic input, the choice of feature and clustering algorithm remains arbitrary. On the other hand, we have the benefit of recording downstream reader neurons, which show that our clustered produced mathematically have physiological correlates: the covariance of CA1 and gRSC spiking state during ripples did not depend on initialization as shown by our parameter sweep which always showed a high fraction of gRSC neurons that discriminated CA1 ripple 'type'.

Reviewer #3 (Remarks to the Author):

If find the authors' rebuttal satisfactory, and in particular I am happy that this revised version clarified the coupling between hippocampal and cortical ripples.

The results are novel and provide important insights regarding the communication between hippocampus and neocortex.

Reviewers' Comments:

Reviewer #1:

Remarks to the Author:

The authors have addressed the core concerns.

Reviewer #2:

Remarks to the Author:

I have read the revised version and found it is improved with new additions. While I still would like to see the direct experiment (closed-loop with no delay) I agree that at least partially the delay experiment helps to address the question, even though specificity of the propagation route is not fully proved.

Although I still disagree with some interpretation and conclusion from the data, this paper reports many interesting observations and I do not want to stand in its way. I feel the research community will benefit from its publication, which I endorse.